# Combatting virulent gut bacteria by inhibiting the biosynthesis of a two-component lanthipeptide toxin

Ryan Moreira [1], Bidisha Chakraborty [2], Yi Yang [1], Chandrashekhar Padhi [1], Michael S. Gilmore [3], Satish K. Nair [1,2,4] ✉ & Wilfred A. van der Donk [1,4] ✉

The enterococcal cytolysin is a toxic, two-component ribosomally synthesized and post-translationally modified peptide (RiPP) produced by pathogenic *Enterococcus faecalis*. Cytolysin-producing (C+) *E. faecalis* resides in the gut microbiome in a commensal role, but results in negative clinical outcomes in alcoholic hepatitis patients. To potentially combat cytolysin virulence, we report inhibitors of its maturation. An extracellular serine protease CylA that is essential for toxin activation is chosen as target. A series of α-aminopeptide boronic acids are designed and synthesized that block cytolysin maturation at low micromolar to nanomolar concentrations in vitro. A crystal structure of CylA provides insights into substrate recognition, autocatalytic activation of the enzyme, and toxin maturation. The inhibitors block hemolytic activity, reduce the amount of cytolysin, and attenuate expression of the cytolysin biosynthetic gene cluster without impeding cell growth. These studies provide a potential route to the development of treatments for cytolysin-induced disease states.

Natural products have had a remarkable impact on human health and many therapeutics are based on natural product scaffolds[1,2]. Notable examples include penicillin[3], streptomycin[4], daptomycin[5], and bleomycin[6] all of which are mainstays in the clinical arsenal for treating life threatening infections and cancers. Outside of the clinic, human health and natural products are also deeply connected as demonstrated by recent advances in microbiome research[7].

The human microbiome consists of trillions of organisms colonising several organ systems throughout the human body[8]. This collection of organisms is responsible for healthy nervous system development[9] and plays important roles in digestion[8] and immunity[10]. Members of the human microbiome communicate with their host and with each other through a number of mechanisms including the production and detection of small molecules[7]. In some cases, these small molecules can significantly impact human health. For example,

colibactin, a natural product produced by *Escherichia coli*, is a genotoxic prodrug that alkylates DNA and has been implicated in colorectal cancers[11,12]. Conversely, lugdunin, a cyclic peptide natural product produced by *Staphylococcus lugdunesis*, mobilizes the human immune system against opportunistic *Staphylococcus aureus*, reducing the incidence and severity of skin infections[13,14].

Ribosomally synthesized and post translationally modified peptides (RiPPs) are a class of natural products in which the substrate sequences are genetically encoded and often clustered together with the genes for the modification enzymes[15]. Lanthipeptides are a subclass of RiPPs that have a wide array of bioactivities ranging from antibacterial to antiviral to morphogenic[16]. When a lanthipeptide has antibiotic properties, it is often referred to as a lantibiotic[17]. The best known lantibiotic is nisin, which has activity against a large panel of Gram-positive organisms and has been used for decades as a food

[1]Department of Chemistry and Howard Hughes Medical Institute, University of Illinois at Urbana-Champaign, Urbana, IL, USA. [2]Center for Biophysics and Quantitative Biology, University of Illinois at Urbana-Champaign, Urbana, IL, USA. [3]Departments of Ophthalmology and Microbiology, Harvard Medical School, Boston, MA, USA. [4]Department of Biochemistry, University of Illinois at Urbana–Champaign, Urbana, IL, USA. ✉e-mail: snair@illinois.edu; vddonk@illinois.edu

preservative against food-borne pathogens[18,19]. Recently, it was discovered that nisin-like lantibiotics are produced by members of the human gut microbiome[20,21]. Furthermore, the presence of these lantibiotics in the gut microbiome decreases the likelihood of colonization of the intestinal tract by vancomycin resistant enterococci (VRE)[20,21]. Another lantibiotic, salivaricin 10, is excreted by commensal organisms found in the mouth and affects the biodiversity of the oral microbiome through immunomodulatory stimulation and selective antimicrobial activity[22]. Conversely, other lanthipeptides produced by members of the human microbiome, such as the enterococcal cytolysin, have the potential to negatively impact human health by complicating existing disease states[23].

The enterococcal cytolysin is a two-component lanthipeptide secreted by *Enterococcus faecalis* that functions as a virulence factor[23]. The lanthipeptide is responsible for hemolytic phenotypes of *E. faecalis* and its role in disease has been known for almost a century[24]. The natural producer of cytolysin is a commensal organism and a member of the human gut microbiome. Its prevalence among humans has not been firmly established, but it is thought that a significant fraction of the human population hosts C+ *E. faecalis*[25]. In

healthy hosts, C+ *E. faecalis* are not known to pose a health risk. In unhealthy hosts, such as those diagnosed with alcoholic hepatitis, the presence of C+ *E. faecalis* is implicated in negative clinical outcomes, resulting in 89% of C+ patients with alcoholic hepatitis dying within 180 days of admission, compared to 3.8% of cytolysin negative patients (C-)[26]. In the same study, it was shown that targeted destruction of the C+ *E. faecalis* using phages attenuated liver toxicity in mouse models. These results suggest that targeted inhibition of the cytolysin biosynthetic pathway could help in treating this deadly disease state.

The biosynthesis of cytolysin has been studied in detail and the process is summarized in Fig. 1. The cytolysin BGC constituted the first reported examples of class II lanthipeptides[27,28]. As with most RiPPs, the gene-encoded substrate is significantly longer than the mature natural product and proteases are involved in the maturation process[29]. The full-length substrate is comprised of two sections: the leader peptide (LP) and the core peptide. The LP is involved in substrate recognition by the biosynthetic enzymes and modulating substrate toxicity[30]. The core peptide undergoes post-translational modification[28].

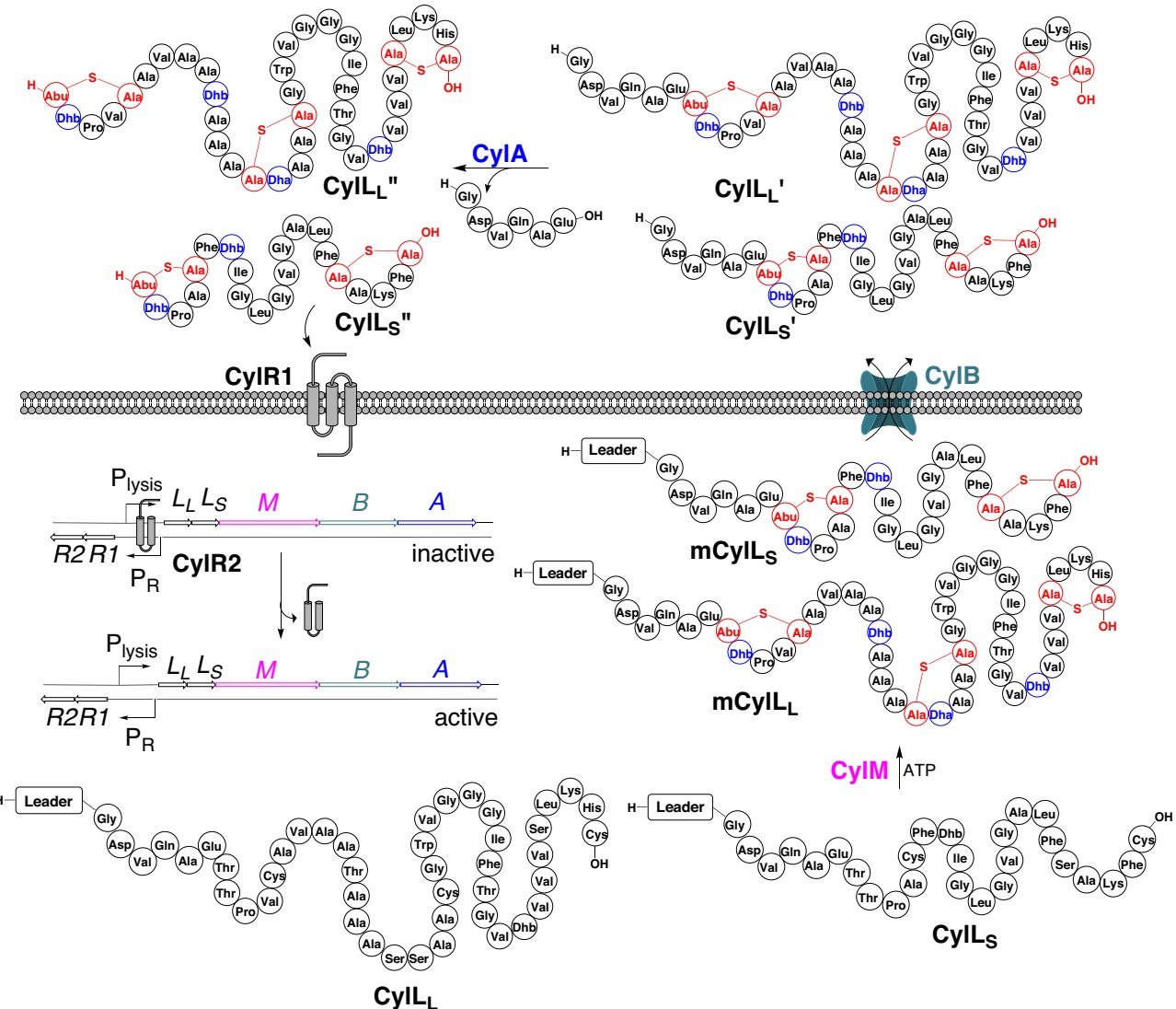

**Fig. 1 | The biosynthesis of cytolysin.** Precursor peptides CylL_L and CylL_S are ribosomally synthesized then modified by CylM producing the polycyclic peptides mCylL_L and mCylL_S. CylB then removes most of the leader peptide and exports CylL_L' and CylL_S' into the extracellular space. An extracellular protease, CylA, removes what remains of the leader peptide from CylL_L' and CylL_S' yielding mature cytolysin (CylL_L" and CylL_S"). CylR1/R2 control cytolysin expression. CylR1 detects free CylL_S" and influences the DNA binding of CylR2 in a manner that increases expression of the cytolysin biosynthetic gene cluster (BGC).

The enterococcal cytolysin BGC contains two precursor genes that encode for CylL$_L$ and CylL$_S$ that are post-translationally modified by a single class II lanthipeptide synthetase called CylM (Fig. 1)[31]. This enzyme dehydrates Thr and Ser residues yielding dehydrobutyrine (Dhb) and dehydroalanine (Dha), respectively, which act as acceptors for enzyme-guided Michael-type additions by Cys side chains forming LL-methyllanthionine (MeLan) and LL- and DL-lanthionine (Lan) bridges[31]. The presence and stereochemistry of these lanthionine residues are important for toxin activity[32,33]. The full-length modified precursors, mCylL$_L$ and mCylL$_S$, are exported and the main part of the LP is removed by a bifunctional transporter-protease, CylB, at GlySer sequences yielding CylL$_L$' and CylL$_S$' (Fig. 1)[34]. An extracellular protease, CylA[35], then cleaves these biosynthetic intermediates yielding the mature natural products CylL$_L$" and CylL$_S$", which cooperatively kill both bacterial and eukaryotic cells at nanomolar concentrations through an unknown mechanism, but which appears to involve membrane disruption[36,37].

Cytolysin is expressed in response to cell density through a quorum sensing mechanism made up of two proteins, CylR1 and CylR2[38,39]. CylR1 is a putative membrane protein that is thought to interact with the transcription factor CylR2[40,41] in a manner that influences DNA binding (Fig. 1). Upon detection of free CylL$_S$", CylR1/CylR2 promotes the transcription of the cytolysin BGC. At low cell densities, cytolysin is expressed at a low level and, in the absence of encroaching cell membranes, CylL$_S$" and CylL$_L$" cooperatively aggregate preventing CylL$_S$" from stimulating CylR1/R2[39]. When the cell density rises, CylL$_L$" is thought to insert into the surrounding cell membranes leaving CylL$_S$" to accumulate in solution. The accumulated CylL$_S$" then interacts with CylR1 in a way that changes the DNA binding of CylR2, thus stimulating cytolysin expression (Fig. 1)[40].

Based on the current knowledge of cytolysin biosynthesis, several enzymes involved in the maturation of cytolysin could be targeted to control cytolysin virulence. Previous studies demonstrated that mCylL$_S$, mCylL$_L$, CylL$_S$', and CylL$_L$' are not hemolytically active[35] and do not stimulate cytolysin expression[38]. Therefore, CylM, CylB, CylA and CylR1/R2 are potential targets to prevent production of active toxin. CylA is a particularly attractive target because of its extracellular localization and its similarity to other proteases that have been successfully targeted by anti-virulence strategies[42,43].

CylA is a subtilisin-like serine protease that is essential for toxin activation[35]. Like many proteases, CylA is translated as a proenzyme that undergoes activation through self-cleavage[44]. Once cleaved, the C-terminal catalytic domain of CylA is thought to non-covalently bind to its pro-domain based on co-purification of these two domains and similarity to a related protease, LicP, which was crystallized with the pro-domain non-covalently bound to the active site[44,45]. The active form of CylA then recognizes the GDVQAE sequence, which it removes from CylL$_L$' and CylL$_S$' to produce the active toxin[35,44]. In RiPP biosynthesis, the C-terminal amino acid in the LP is numbered −1 and other residues in the LP are numbered backwards from this position with negative numbers. The putative self-cleavage recognition sequence (PDFKPE) and the substrate recognition sequence (GDVQAE) are different but do share Glu and Asp at positions −1 and −5, respectively. Previously, it was suggested that Glu−1 is necessary for positioning the substrate in the enzyme active site through an interaction with the side chain of His275 – a His that flanks the active site[35]. Although in vivo LP removal is a two-step process involving first CylB and then CylA, in vitro CylA can remove the entire LP in one step[44]. Despite limited details describing the interaction of CylA with its self-cleavage or substrate sequence, existing models suggest that these sequences may serve as starting points for the development of CylA inhibitors.

In this work, a series of peptide boronic acid inhibitors of CylA was developed based on its natural recognition sequences, and this collection of inhibitors was used to gain insights into the interaction of CylA with peptidic substrates. In vitro studies demonstrate that peptide boronic acids with sequences similar to the substrate or self-cleavage sequence inhibited the proteolytic activity of CylA and blocked the conversion of mCylL$_L$ and mCylL$_S$ to their mature and toxic counterparts. Studies on enzyme-inhibitor interactions reveal that CylA undergoes activation in a substrate-dependent manner and that inhibitor binding facilitates irreversible release of the pro-domain from the catalytic C-terminal domain. An X-ray structure of CylA non-covalently bound to its pro-domain provided a high-resolution glimpse into substrate-enzyme interactions. Finally, experiments on C+ *E. faecalis* demonstrate that these inhibitors attenuated the hemolytic ability of cytolysin producers by halting the maturation of cytolysin and the expression of the cytolysin BGC.

## Results and discussion
### Synthesis of peptide boronic acid inhibitors for CylA
Boronic acids are attractive warheads for inhibitors of serine proteases and several FDA approved drugs contain boronic acids[46]. CylA is a subtilisin-like serine protease possessing a Ser-Asp-His catalytic triad[44] and proteases of this type have been targeted by boronic acid containing peptides leading to beneficial decreases in enzyme activity in vivo[42]. CylA cleaves after the N-terminal hexapeptide sequence GDVQAE, and probably recognizes the PDFKPE sequence at its self-cleavage site[44]. We envisioned replacing the carboxylic acid terminus of each hexapeptide sequence with a boronic acid to generate inhibitors that bind covalently but reversibly to CylA preventing conversion of CylL$_S$' and CylL$_L$' to their mature counterparts.

Boronic acid-containing hexapeptide inhibitors were synthesized through a convergent approach detailed in Fig. 2. The pentapeptide fragments were prepared using Fmoc solid phase peptide synthesis (SPPS) starting from 2-chlorotrityl (2ClTrt) resin loaded with Fmoc-L-Ala, Fmoc-L-Abu, or Fmoc-L-Pro. Standard Fmoc SPPS conditions were used to extend to the −6 position, which was acetylated yielding peptides **1a-9a**. Each peptide was removed from the solid support with 1% trifluoroacetic acid (TFA) keeping all the side chain protecting groups intact yielding protected peptides **1b-9b**. The side chains of **1a-9a** and **1b-9b** match the compound with the same number presented in Table 1; for example, **1a** was the precursor to **1b** and **1b** was the precursor to **1**. α-Amino boronic esters were prepared enantioselectively starting from aldehydes **11** and **12**, which were converted to *N*-*tert*-butanesulfinyl imines **13** and **14**, respectively, through treatment with CuSO$_4$ in the presence of (*R*)-(+)-2-methyl-2-propanesulfinamide[47,48]. Compounds **13** and **14** were subjected to diastereoselective borylation yielding α-sulfinamide boronic esters **15** and **16**, respectively[49]. Removal of the sulfinamide from **15** and **16** occurred readily yielding **17** and **18**[49], respectively, which were immediately coupled to **1b-9b** using 1-[bis(dimethylamino)methylene]-1H-1,2,3-triazolo[4,5-b]pyridinium 3-oxid hexafluorophosphate (HATU) and *N*,*N*-diisopropylethylamine (DIPEA). Following aqueous workup, the crude protected peptides were globally deprotected by treatment with TFA/H$_2$O/triisopropylsilane (TIPS), then precipitated from cold diethyl ether and purified by ultra-high-performance liquid chromatography (UHPLC) yielding **1-9** (Supplementary Figs. 1-2). During treatment of **9b** with TFA/TIPS/H$_2$O, up to 30% of the deprotected peptide underwent protodeboronation, giving peptide **10**, which was isolated and used to assess the impact of the boronic acid warhead in vitro. The partial protodeboronation of these peptides is not suggestive of instability; alkylboronic acids are known to undergo protodeboronation under the strongly acidic conditions used for deprotection but are stable at physiological pH[50,51].

### Peptide boronic acids are potent inhibitors of His$_6$-CylA-27-412 in vitro
We first tested the ability of compound **1**, which contains the natural substrate recognition sequence, to inhibit the CylA-catalyzed cleavage

**Fig. 2 | Synthesis of compounds 1-10.**

of mCylL$_S$ and mCylL$_L$ in vitro using liquid chromatography with mass spectrometric detection (LCMS) (Fig. 3). mCylL$_S$ and mCylL$_L$ were used in place of the natural substrates, CylL$_S$' and CylL$_L$', because they are known to be accepted by CylA, have enhanced solubility in aqueous solution, and are more accessible as they do not require the membrane-bound protease CylB[33,44]. We expressed CylA in *Escherichia coli* with an N-terminal His$_6$-tag and without the secretion signal that is present in WT CylA. After purification by metal affinity chromatography, incubation of each substrate (10 µM) with this purified His$_6$-CylA-27-412[44] (4 nM) gave partial conversion to the mature core peptide after 30 min at room temperature. Addition of compound **1** (1 µM) completely blocked His$_6$-CylA-27-412-catalyzed cleavage over the 30 min reaction time (Fig. 3).

Cytolysin causes hemolysis and lyses a variety of eukaryotic cells[36,37]. Additionally, CylA aggregates and is thought to associate primarily with the membrane of the producing cell[52,53]. To mimic the maturation of cytolysin in vitro, an enzyme assay was designed that linked toxin production to the lysis of fluorescently labeled liposomes. In this assay, mCylL$_L$ and mCylL$_S$ were combined in a 1:1 ratio and treated with His$_6$-CylA-27-412 in the presence of pyranine-containing liposomes in buffer of pH 7.4 (Fig. 4A). Within the liposomes was a buffer of lower pH (6.0), and therefore any pore formation or membrane disruption resulted in an increase in internal pH, which increased the fluorescence signal of the pyranine. In this way, pyranine fluorescence was correlated to toxin activation allowing for the determination of EC$_{50}$ values (Fig. 4B, C). Using this approach, the inhibitory activities of compounds **1-10** were evaluated (Table 1).

Compound **1**, which mimics the natural recognition sequence of the substrate, inhibited liposome lysis with an EC$_{50}$ of $30 \pm 3$ nM. Peptide **2**, which is based on the self-cleavage sequence of CylA,

showed an EC$_{50}$ of $117 \pm 16$ nM, about 4-fold higher than **1**. Comparing the activity of **1** and **3**, replacing the C-terminal Glu-mimicking boronate with a norvaline-mimicking boronate did not significantly change the EC$_{50}$. This observation was unexpected since both the substrate and self-cleavage sequence contain Glu−1[44], the related enzyme LicP requires Glu at −1 of its substrate[45], and an important salt bridge between the side chain carboxyl group of Glu−1 and His275 of CylA has been proposed[35]. Replacement of Asp−5 in **1** with a Glu residue (**4**) resulted in a 25-fold increase in EC$_{50}$ and replacement of the same Asp with 2-aminobutyric acid (Abu, **5**) increased the EC$_{50}$ 224-fold. These two results suggest that Asp−5 is involved in an important interaction with the enzyme, consistent with the substrate interaction model proposed for LicP[45], which contains a salt bridge between Asp−5 and a proximal Lys residue. Replacement of Ala−2 in compound **1** with Abu (**6**) resulted in a ~ 2-fold increase in EC$_{50}$. Switching Gln−3 to Asn (**7**) marginally increased the EC$_{50}$ and replacement of Gly−6 with Ser (**8**) produced a similar increase suggesting that some variability is tolerated at these positions. We hypothesized that converting Val to cyclopropylglycine (Cpg) could increase inhibitor effectiveness through conformational constriction of the side chain and a potential entropic advantage. Indeed compound **9** had an EC$_{50}$ of $12.5 \pm 1.4$ nM, which is 2-fold more effective than compound **1** at inhibiting pore formation. The protodeboronated form of **9**, compound **10**, also inhibited pore formation but it was 115-fold less effective than boronic acid containing **9**.

## Substrate and inhibitor activation of CylA activity at low concentrations

A FRET assay was developed to directly detect the inhibition of His$_6$-CylA-27-412 activity by inhibitor **9**. Fmoc SPPS was used to prepare a

**Table 1 | Structures and EC$_{50}$ values of inhibitors 1-10**

| Compound | Structure | EC$_{50}$ (nM) |
|---|---|---|
| 1 | | $30 \pm 3$ |
| 2 | | $117 \pm 16$ |
| 3 | | $35 \pm 6$ |
| 4 | | $748 \pm 80$ |
| 5 | | $(6.7 \pm 0.8) \times 10^3$ |
| 6 | | $70 \pm 11$ |
| 7 | | $37 \pm 3$ |
| 8 | | $40 \pm 3$ |
| 9 | | $12.5 \pm 1.4$ |
| 10 | | $(1.4 \pm 0.3) \times 10^3$ |

Errors represent the error in the non-linear concatenate fit of two separate trials ($n = 2$).

FRET labeled substrate. The substrate recognition sequence and a portion of the N-terminus of unmodified CylL$_S$ were embedded in between N-terminal Asp(EDANS) and C-terminal Lys(DABCYL) residues (Fig. 5A). The FRET labeled substrate was incubated with nanomolar concentrations of His$_6$-CylA-27-412 at 37 °C overnight. High resolution LCMS-MS analysis revealed that the peptide was cleaved under these conditions producing fragments consistent with cleavage at the C-terminus of Glu (Supplementary Figs. 4 and 5). Visually, a sample of the FRET peptide treated with His$_6$-CylA-27-412 was significantly more fluorescent than a CylA-free control (Fig. 5A) showing that the peptide worked as designed.

The FRET peptide was used to continuously monitor the activity of His$_6$-CylA-27-412 at different substrate concentrations. Plotting the changes in fluorescence emission intensity as a function of total substrate concentration yielded a sigmoidal curve indicative of substrate activation (Fig. 5B). Fitting this curve to the Hill form of the Michaelis-Menten equation gave a $K_m$ equal to $18.3 \pm 1.4$ μM and a Hill co-efficient (h) of $1.8 \pm 0.2$, suggesting significant activation of CylA by the FRET peptide substrate. Treating samples of His$_6$-CylA-27-412 with mCylL$_S$, mCylL$_L$ and/or a mixture of mCylL$_S$:mCylL$_L$ increased the initial rate of FRET peptide cleavage by a factor of up to $1.5 \pm 0.3$ (Supplementary Fig. 6) at total peptide concentrations in the nanomolar range but above the total concentration of CylA. In contrast, micromolar concentrations of pre-cytolysin peptides lowered the rate of FRET peptide proteolysis suggesting that the pre-cytolysin peptides are competing for the same active site as the FRET substrate under these conditions.

Treating His$_6$-CylA-27-412 with different concentrations of compound **9** showed inhibition with an IC$_{50}$ of $8.7 \pm 0.9$ nM (Fig. 5C). Similar to the increase in CylA activity at low concentrations of pre-cytolysin peptides, compound **9** activated His$_6$-CylA-27-412 at sub-stoichiometric concentrations (the total enzyme concentration was 20 nM). When His$_6$-CylA-27-412 was treated with 762, 254, or 85 pM of **9**, the rate of FRET peptide cleavage increased significantly. These data suggest that activation of His$_6$-CylA-27-412 occurs under conditions where inhibitor binding is competent, but active site occupancy is low.

Substoichiometric activation of a serine protease by a peptidic boronic acid inhibitor based on the natural substrate sequence of the enzyme has been characterized for DegP and HTRA1, which are conformation-specific protease/chaperones[54]. Detailed biophysical studies on the activation of DegP by substoichiometric quantities of peptidic boronic acid inhibitor revealed that binding of the inhibitor to the active site of a protomer increased the activity of adjacent protomers in the resting hexameric state, while stoichiometric quantities of the inhibitor produced an inactive dodecameric form. Both DegP and HTRA1 are S1 proteases that differ structurally from CylA (an S9 protease), raising questions about the mechanism of substoichiometric activation of CylA by compound **9**. To our knowledge, substrate activation and substoichiometric activation by a substrate-based boronic inhibitor have not been reported for S9 subtilisins[42]. Enzymes of this class undergo maturation through a general mechanism in which the rate determining step is often dissociation of the N-terminal pro-domain from the enzyme active site[55]. It is possible that the release of the CylA pro-domain from the C-terminal catalytic domain is accelerated by substrate binding, or by binding substrate-like species, like inhibitor **9**. If the pro-domain is slow to reassociate with the C-terminal catalytic domain due to unfolding upon displacement[55], an increase in CylA activity could be observed at inhibitor concentrations that rapidly equilibrate with the enzyme active site, but give marginal active site occupancy[55]. Alternatively or simultaneously, the increase in activity could follow a DegP-like mechanism in which inhibitor binding influences oligomeric states.

To gain more insight into this phenomenon, His$_6$-CylA-27-412 that had undergone complete autolytic cleavage as determined by SDS-PAGE and that had been purified by size exclusion chromatography (SEC), was analyzed by analytical SEC (aSEC) in the

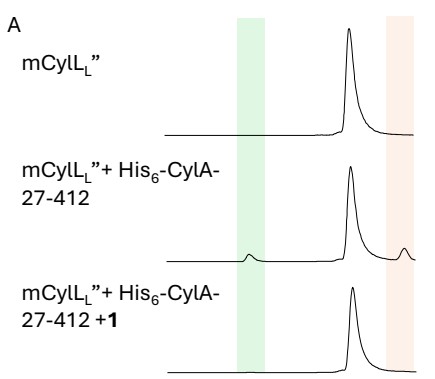

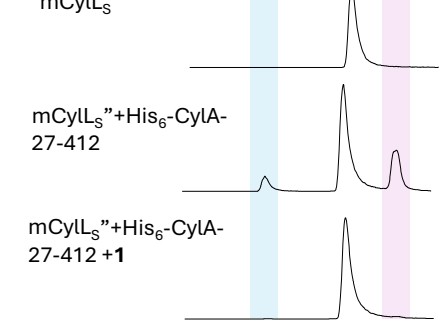

**Fig. 3 | Inhibitor 1 (1 µM) blocks the His$_6$-CylA-27-412 (4 nM) catalyzed cleavage of mCylL$_L$ and mCylL$_S$ (10 µM) during 30 min incubation, preventing the formation of CylL$_L$" and CylL$_S$", respectively. A** Peak highlighted in green is the LP and peak highlighted in orange is CylL$_L$". **B** Peak highlighted in blue is the LP and peak highlighted in violet is CylL$_S$". Longer incubation times result in near-complete cleavage of the substrates[33]. Source data are provided as a Source Data file.

presence of inhibitor **9**. The aSEC trace showed two peaks, which had retention times that corresponded to masses consistent with CylA-96-412 with and without its pro-domain non-covalently attached (Fig. 5D and Supplementary Fig. 7). This assignment was supported by SDS-PAGE analysis of the eluted fractions (Supplementary Fig. 8). The addition of substoichiometric quantities of compound **9** (1:8 **9**:CylA) increased the complexity of the trace in the region where CylA-96-412 and CylA-96-412 non-covalently bound to its pro-domain (CylA-96-412° His$_6$-27-95) elute when **9** was not present. Although several overlapping peaks seem to have formed, two pronounced peaks on each end of the 10.5-12.5 min grouping were observed (Fig. 5D). The peak eluting at 12.1 min had a retention time that was slightly longer than CylA-96-412 (11.9 min) and the peak eluting at 11.3 min had a retention time that was slightly shorter than CylA-96-412°His$_6$-27-95 (11.4 min). Adding more inhibitor relative to enzyme increased the intensity of the peak at 12.1 min relative to the CylA-96-412°His$_6$-27-95 signal and the transition appeared complete once a stoichiometric equivalent of inhibitor was added (Fig. 5D). To see if pro-domain ejection occurred in a substrate-dependent manner, a similarly prepared sample of His$_6$-CylA-27-412 was treated with mCylL$_S$, mCylL$_L$ or a 1:1 mixture of mCylL$_S$ and mCylL$_L$ for 10 min at room temperature and then analyzed by aSEC. In each case, a relative decrease in the amount of CylA-96-412°His$_6$-27-95 was observed (Supplementary Fig. 9). The SEC retention time data is inconsistent with CylA forming higher order oligomers, ruling out a DegP-like activation mechanism. Instead, the SEC data is consistent with inhibitor or substrate binding to both CylA-96-412 and CylA-96-412°His$_6$-27-95 and encouraging but not necessitating pro-domain displacement from CylA-96-412°His$_6$-27-95. To assess whether pro-domain displacement is reversible, the isolated pro-domain was incubated with a sample of SEC-purified His$_6$-CylA-27-412 for 20 min at room temperature and then analyzed by aSEC (Fig. 5E). No change in the ratio of CylA-96-412:CylA-96-412°His$_6$-27-95 was observed demonstrating that once the pro-domain is released from the C-terminal domain it cannot readily rebind. Analysis of the pro-domain isolated during the SEC purification of His$_6$-CylA-27-412 using circular dichroism (CD) showed it was in an unfolded state (Supplementary Fig. 10). These results are consistent with pro-domain folding being essential for and dependent on interactions with the C-terminal domain, which is typical for related proteases[55].

Collectively, these observations are consistent with binding of inhibitor or substrate encouraging irreversible release of the pro-domain resulting in an increase in the population of CylA-96-412 relative to CylA-96-412°His$_6$-27-95 and increasing enzyme activity. The increase in enzyme activity was small (1.3-1.5-fold) in most cases probably because CylA-96-412 was already present in the sample and CylA-96-412°His$_6$-27-95, which can bind to the inhibitor, may have some proteolytic activity. Inhibitor and substrate activation suggests that CylA may release its pro-domain more rapidly when natural substrate concentrations begin to increase. This mechanism may protect the natural producer from the proteolytic CylA until substrate concentration surpasses a threshold level. Such a mechanism may provide another point of control for cytolysin biosynthesis and prevent triggering of the CylR1/CylR2 response at lower cell densities when substrate expression is at basal levels[39]. Importantly, the dose-response studies (Fig. 5C) demonstrate that inhibitor **9** can inhibit CylA at low nanomolar concentrations.

## The structure of the CylA pro-domain-catalytic domain complex

To afford a high-resolution structure, full-length CylA (residues Leu27-Lys412, lacking the N-terminal secretion signal sequence) with an N-terminal polyhistidine tag was expressed in *E. coli*, refolded from the insoluble fraction and incubated in situ to allow for autoproteolytic processing. Following near-complete auto-processing, as determined by SDS-PAGE analysis, the soluble, folded CylA-96-412°His$_6$-27-95 complex was purified by nickel affinity chromatography. The cleaved domains co-eluted as a single species suggesting that the pro-domain was tightly bound to the protease domain. Following size-exclusion chromatography, diffraction quality crystals of the complex were obtained using the vapor diffusion method. Crystals of CylA-96-412°His$_6$-27-95 diffracted to a Bragg limit of 1.3 Å and initial crystallographic phases were determined using an AlphaFold[56] model of the catalytic domain as a molecular replacement search probe. The -75-residue pro-domain was built using a combination of automated and manual methods into electron density maps calculated using initial phases calculated from the catalytic domain. A superposition of the final experimental structure with the AlphaFold predicted structure showed deviations in the orientation of the pro-domain and catalytic domain likely due to constraints between the two domains in the predicted structure (Supplementary Fig. 11).

The crystal structure of CylA-96-412°His$_6$-27-95 reveals the complex between the catalytic domain and the proteolytically excised pro-domain (Fig. 6A). The active site of the catalytic domain, consisting of the catalytic triad residues Asp142, His173, and Ser359, is additionally occupied by the C-terminal tail of the pro-domain ending with the α-carboxylate of residue Glu95 (position −1 for autoproteolysis), which is located 3.2 Å away from the nucleophilic Ser359 side chain oxygen. The side chain of Asn263 and the backbone amide of Thr275 are positioned to provide an oxyanion hole to stabilize the negative charge

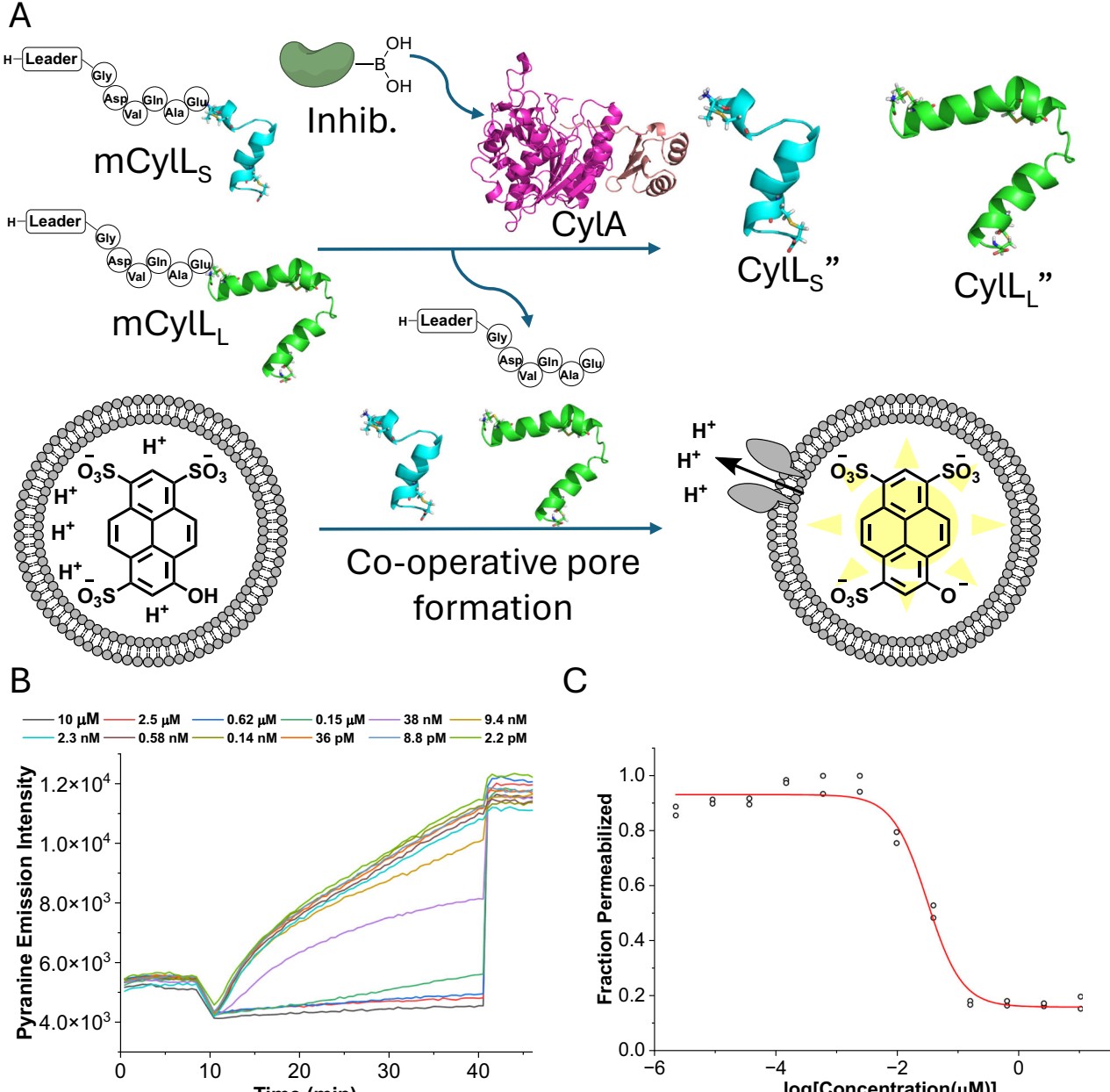

**Fig. 4 | In vitro pore formation assay for determining inhibitor effectiveness.**
**A** A schematic of the assay showing how pore formation is used as an indicator of toxin maturation. His₆-CylA-27-412 cleaves mCylL_S and mCylL_L producing CylL_S″ and CylL_L″, which cooperatively form pores in pyranine-encapsulated DOPC liposomes at nanomolar concentrations. Pore formation dissipates a pH gradient between the interior of the liposome and the surrounding buffer increasing pyranine fluorescence. **B** Representative example of changes in membrane

permeabilization activity with decreasing concentration of **1**. **C** Representative plot of the permeabilization activity as a function of the concentration of **1** fit to a dose-response curve. Fitting was done using Origin Pro 2024 yielding EC₅₀ values (see Supplementary Fig. 3 for additional data for compounds **2**-**10**). The data points from two replicates are shown (*n* = 2). Source data are provided as a Source Data file.

accumulated in the transition state. Clear and continuous electron density is evident for the entirety of the pro-domain tail detailing interactions with the PDFKPE recognition sequence preceding the scissile bond. The pro-domain and the catalytic domain form extensive contacts that orient the recognition sequence into the active site of the latter (Supplementary Fig. 12). Formation of the interface buries ~1270 Å² of surface area as determined using the Proteins, Interfaces, Structures and Assemblies (PISA) server[57]. Pairwise comparison of the CylA protease domain against the Protein Data Bank, using the DALI server, shows an overall conservation with other subtilisin-like lanthipeptide proteases such as LicP[45] (PDB Code 4ZOQ; Z-score=42.9,

RMSD of 2.0 Å over 302 aligned Cα residues), the isolated protease domain NisP[58] (PDB Code 4MZD; Z-score=35.6, RMSD of 2.3 Å over 280 aligned Cα residues), and the as of yet uncharacterized homolog of EpiP[59] (PDB Code 3QFH; Z-score=35.5, RMSD of 2.4 Å over 275 aligned Cα residues). The CylA structure is also related to other canonical S9 proteases, including various homologs of subtilisin.

A closer examination of the active site shows interactions with the pro-domain PDFKPE C-terminal sequence that account for substrate specificity (Fig. 6B, Supplementary Fig. 13). The side chain of Glu95 at the −1 position is within hydrogen bonding distance of the side chains of Ser260, His275 and the backbone amide of Gly262. The

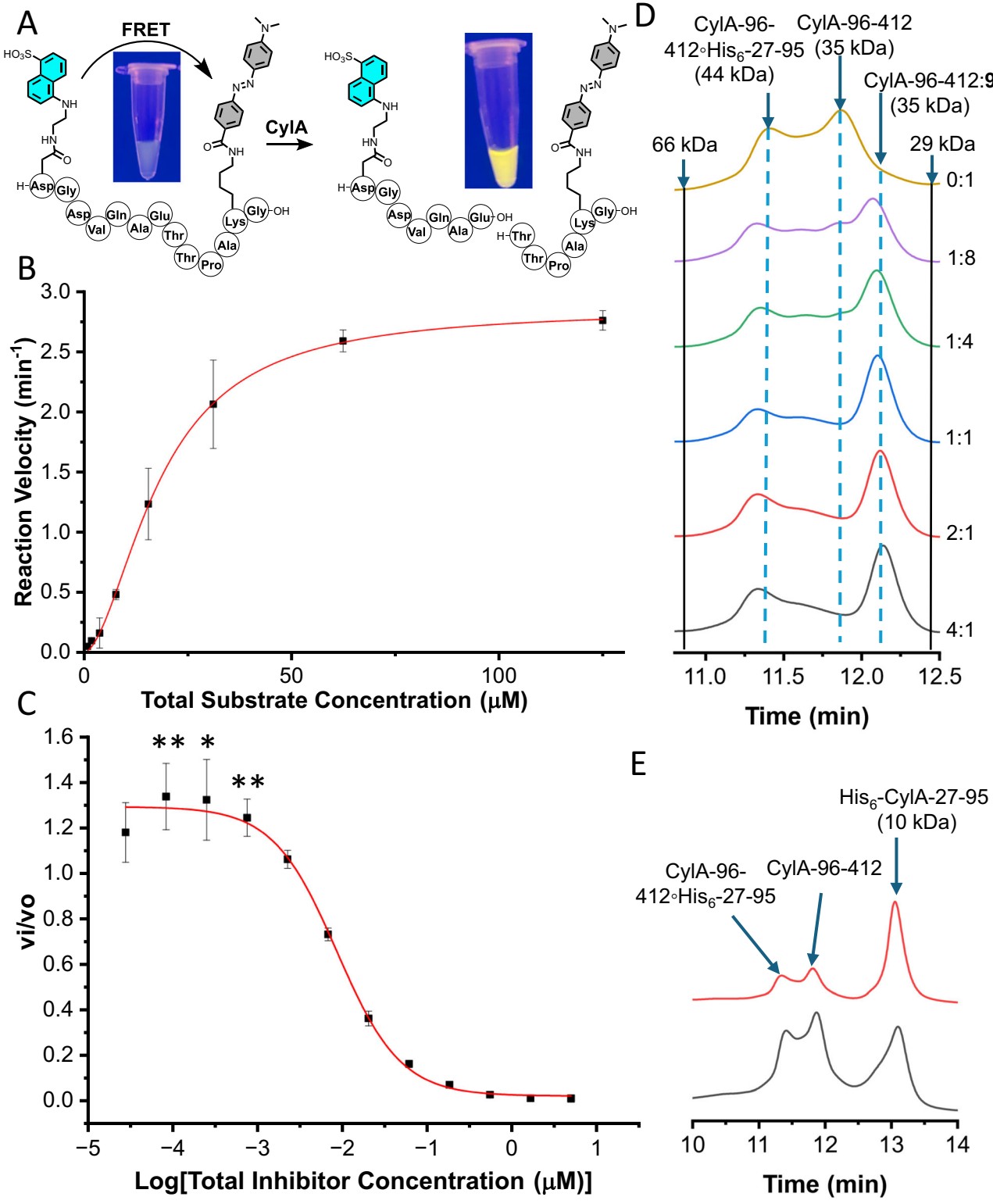

sidechain of Glu264 is outside hydrogen bonding distance to Glu95 (4.4 Å), but may be involved in electrostatic stabilization of protonated His275. The side chain of the Ser359 catalytic nucleophile in the X-ray structure is well positioned to engage the boronic acid of the inhibitors in a model (Supplementary Fig. 14). The residues at positions −2 (Pro94) and −3 (Lys93) face away from the catalytic domain, accounting for the lack of conservation between the pro-domain cleavage site and the GDVQAE sequence in the $CylL_L'$ and $CylL_S'$ substrate peptides. Notably, the −4 residue (Phe92) is directed into

the catalytic domain where it is engaged in van der Waals contacts with residues in a pocket composed of Val195, Met196, Met206, and Leu225. These interactions provide a rationale for the preference of a hydrophobic residue (Val or Phe) at the −4 position in both the self-cleavage and in the substrate peptide of CylA, as well as in the related LicP protease. Lastly, Asp91 at the −5 position is directed away from the catalytic domain but is located within hydrogen bonding distance of Asp200 and Asn202. These observations suggest that CylA should show higher specificity at −1 and lower specificity at the −5 position,

**Fig. 5 | FRET assay used to continuously monitor His₆-CylA-27-412 activity and aSEC data describing the process of inhibitor-induced pro-domain release.**
**A** Structure of the FRET substrate and its cleavage by CylA. Samples of FRET substrate (44 μM) in PBS (pH = 7.4) before and after treatment with His₆-CylA-27-412 (200 nM) overnight at 37 °C. **B** Initial rates of fluorescent peptide formation as a function of initial FRET substrate concentration (2–125 μM). Non-linear concatenate fitting of data from three independent replicates ($n = 3$) to the Hill form of the Michaelis-Menten model using Origin Pro 2024 gave $K_m$, h and associated errors. Points represent the mean. Error bars represent the standard of deviation and are centered at the mean. **C** The relationship between relative enzyme activity ($v_i/v_o$) and concentration of **9**. Relative enzyme activity was determined by monitoring the rate of cleavage of the FRET peptide in the presence of inhibitor ($v_i$) and comparing the rate to an uninhibited control ($v_o$). Non-linear concatenate fitting of data points $v_i/v_o$ from four independent replicates ($n = 4$) to a three-parameter function was used to determine IC₅₀. The colored line represents the calculated fit function. Points represent the mean and error bars represent the standard of deviation centered at the mean. **D** aSEC analysis of SEC-purified His₆-CylA-27-412 treated with different concentrations of inhibitor **9**. The inhibitor:CylA ratio is indicated next to each trace. **E** aSEC traces of SEC-purified His₆-CylA-27-412 before (black) and after (red) incubation with purified samples of pro-domain. For (**D**) and (**E**) absorbance at 220 nm was monitored. Full length versions of the traces featured in (**D**) and (**E**) can be found in Supplementary Fig. 7. Statistical significance was calculated using a one sample and one-sided $t$ test. *$P < 0.05$, **$P < 0.01$. Source data are provided as a Source Data file.

which contradicts the data presented in Table 1, and suggests that the substrate recognition site of CylA undergoes some restructuring upon release of the pro-domain and/or upon substrate binding.

Despite much effort, we were not able to obtain a structure of an inhibitor bound to the catalytic domain. To gain more insight into the hypothesized restructuring, we used AlphaFold 3[60] to generate a structure of CylA-96-412 with GDVQAE bound to the substrate recognition site (Supplementary Fig. 15). The structure shows GDVQAE bound to the substrate recognition groove near the active site and CylA-96-412 aligns well with the crystal structure (RMSD = 0.5 Å over 386 aligned Cα residues); however, deviations are concentrated at a location near the substrate recognition groove. Notably, in the AlphaFold 3 structure, a short, anti-parallel β-sheet located near the substrate binding groove orients the side chain of Lys229 towards Asp −5 generating a salt bridge. In the crystal structure, this region does not contain the anti-parallel β-sheet and the Lys229 ε-NH₂ is 12.4 Å from the Asp−5 carboxylate. These observations suggest that this region of CylA may be dynamic, undergoing spontaneous re-structuring in solution or in response to pro-domain release and/or substrate binding. The latter hypothesis is consistent with the observed activation of CylA when it is treated with substrate or substoichiometric concentrations of substrate mimicking inhibitor.

A structure-based DALI search against the PDB using the coordinates of only the pro-domain revealed structural similarities that may be instructive. As expected, the CylA pro-domain (Fig. 7A) shares structural similarity with the pro-domain of subtilisin BPN' (PDB Code 1SPB, Z-score = 3.9, RMSD of 3.3 Å over 58 aligned Cα residues)[61]. However, structural similarity is also detected with the RiPP recognition element (RRE) domain of NisB (PDB Code 4WD9, Z-score=4.6, RMSD of 3.0 Å over 62 aligned Cα residues; Fig. 7B)[62]. Weaker structural resemblance is also noted between the CylA pro-domain, and the nitrile hydratase-like protein (NHLP) type LPs found in proteusin biosynthetic pathways (Fig. 7C; see Supplementary Fig. 16 for overlay of all three domains)[63]. The RRE, NHLP and the protease pro-domains interact with and stabilize their cognate catalytic domain in the active form. Notably, cytolysin is a product of a class II lanthipeptide pathway and none of the biosynthetic enzymes employ either an RRE or a NHLP peptide. Hence, the structural similarity of the CylA pro-domain with domains involved in the biosynthesis of other classes of RiPPs represents a convergent strategy for engaging catalytic domains with their substrates.

### Compound 9 inhibits toxin maturation in C+ *E. faecalis*

Inhibitor **9** showed promising activity in vitro, suggesting that it may effectively prevent toxin production in C+ *E. faecalis*. To test this hypothesis, a solution-phase hemolysis assay was developed, and this assay was used to monitor the effect of **9** on the hemolytic activity of several strains of *E. faecalis* (Fig. 8A). The cytolysin BGC is part of conjugative plasmid pAD1[64]. *E. faecalis* FA2-2 (-pAD1) lacks pAD1 and *E. faecalis* FA2-2 (pAM9055) is defective in *cylA*[27,64]. Consistent with previous findings, both strains were not hemolytic (Fig. 8A). *E. faecalis* FA2-2 (pAM714) possesses an intact cytolysin BGC[64] and was found to

be hemolytic. Treating *E. faecalis* FA2-2 (pAM714) with **9** significantly reduced hemolytic activity relative to untreated cells. Repeating this experiment on C+ *E. faecalis* ATCC 29212, showed that treatment with **9** again significantly reduced hemolysis. This assay was conducted over a range of inhibitor concentrations and an EC₅₀ equal to $52 \pm 8$ μM was determined (Fig. 8B). Attenuation of hemolytic activity was not caused by toxicity of **9** to *E. faecalis* FA2-2 (pAM714) or ATCC 29212 because **9** did not impact the growth of C+ *E. faecalis* at 100 μM (Supplementary Fig. 17). Furthermore, toxicity studies involving HeLa cells found inhibitors **2, 3,** and **5-9** to be non-toxic at concentrations ≤ 100 μM (Supplementary Fig. 18) and **9** displayed relatively good stability in human serum (Supplementary Fig. 19A) suggesting protease selectivity and potential utility for in vivo studies.

To determine if inhibitor **9** blocks the maturation of cytolysin in C+ *E. faecalis*, an LCMS method was developed to estimate the concentration of CylL_S″ in cell extracts. This method employed biosynthetically prepared ¹⁵N,¹³C-labeled CylL_S″ as an internal standard[65]. Cultures of *E. faecalis* were treated with inhibitor **9** or vehicle then lyophilized and extracted with methanol. Cell samples treated with 1 or 10 μM of **9** produced similar or significantly more CylL_S″ relative to samples that were not treated with inhibitor **9** (Fig. 8C). In contrast, cells treated with 100 μM inhibitor **9** produced quantities of CylL_S″ that were on average 5.5-fold less than the untreated control. These experiments demonstrated that inhibitor **9** can slow the production of cytolysin in dense cultures that are manufacturing the toxin over many hours. Inhibitors of the agr system that control production of toxin virulence factors in *Staphylococcus aureus* showed a similar activity profile of higher in vitro potency than in cellular settings, but nevertheless proved effective in mouse models[66].

The increase in the amount of extractable CylL_S″ observed in samples treated with 1 or 10 μM inhibitor **9** compared to untreated samples is intriguing (Fig. 8C). Based on the data presented in Fig. 5C-E and Supplementary Fig. 6, it is tempting to conclude that these increases are due to the activation of CylA by inhibitor **9**. However, other explanations can be surmised. For example, it is known that CylL_S′ is a better substrate for CylA than CylL_L′ and thus imbalances favoring CylL_S″ maybe be observed when CylA is partially inhibited[44]. These imbalances may be compounded by the co-operative aggregation of CylL_S″ and CylL_L″ resulting in increases in extractable CylL_S″ that are not reflective of the rate of toxin maturation.

To investigate the stability of **9** in the presence of *E. faecalis*, cell extracts treated with **9** were tested for inhibitory activity. Extracts of overnight cultures that were not treated with **9** did not impact the activity of CylA in vitro (Supplementary Fig. 19B). Cultures that were treated with 100 μM of **9** at 37 °C for 20 h gave extracts that inhibited the activity of CylA in vitro. The inhibitory activity of these extracts was about 3-fold less than a sample of fresh **9**. These results suggest that **9** has relatively good stability in cultures of *E. faecalis*.

Finally, to determine if CylA inhibition resulted in a reduction of the transcription of the cytolysin BGC, RT-qPCR was performed on samples of total RNA extracted from overnight cultures of C+ *E. faecalis*. Since the cytolysin BGC is often expressed in two transcripts

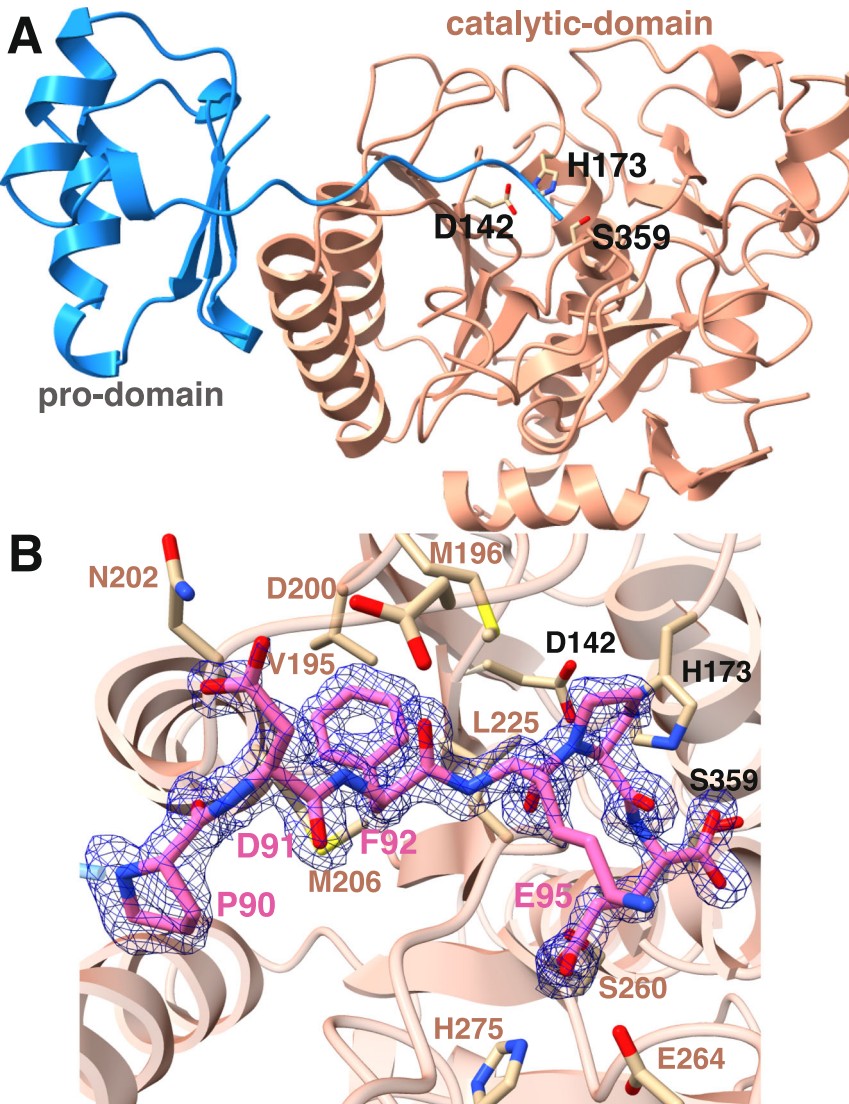

**Fig. 6 | Structure of CylA-96-412°His₆-27-95. A** Overall structure showing the orientation of the pro-domain and the catalytic domain. The catalytic triad residues are shown in a stick representation. **B** Interactions between the C-terminal residues of the pro-domain (PDFKPE) and CylA-96-412. Data deposited in PDB: 9EKQ. The C-terminal carboxylate of the pro-domain is in close proximity of the hydroxy side chain of the catalytic nucleophile Ser 359. A difference Fourier omit map (Fo-Fc) calculated with the coordinates of the cleavage peptide omitted prior to refinement is superimposed and shown at a contour level of 3σ (blue).

because of the presence of a weak intergenic terminator between *cylL_L* and *cylM*, primers and amplicons were selected for *cylL_L*, *cylL_S* and *cylA*[67]. Compared to an untreated control, treatment of *E. faecalis* with **9** (100 µM) resulted in an 86-fold reduction in *cylA* transcripts and a 9-10 fold reduction in *cylL_L* and *cylL_S* transcripts (Fig. 8D). These results demonstrated that, in addition to preventing toxin maturation, CylA inhibition strongly reduces the transcription of the cytolysin BGC, as predicted by the model in Fig. 1.

In closing, the intersection between the human microbiome, human health and natural products has added an exciting dimension to natural products research. At this interface, the biosynthetic information amassed from extensive study of natural products biosynthesis finds application that can directly impact human health. In this work, knowledge of the cytolysin BGC was used to rationally develop small molecule inhibitors against this deadly virulence factor, which directly contributes to high mortality in patients that host C+ *E. faecalis* and have alcoholic hepatitis[26]. Targeting a key protease that is essential for toxin activation, substrate-based inhibitors were developed that had low micromolar to low nanomolar activities depending on the experimental context. Studies on the inhibition of CylA using these inhibitors revealed that CylA is a substrate-activated protease and this feature provides an additional potential point of control for cytolysin production in natural settings. A crystal structure of the CylA catalytic domain non-covalently bound to its pro-domain, in combination with inhibitor-enzyme structure-activity studies, provided a detailed description of substrate recognition. Finally, bioactivity studies demonstrated that inhibitors of CylA blocked the hemolytic activity of C+ *E. faecalis* by preventing toxin maturation and attenuating expression of the cytolysin BGC.

## Methods
### General methods
All reagents used for chemical synthesis were purchased from Sigma Aldrich, Chem-Impex and CombiBlocks. *N,N*-Dimethylformamide (DMF) and dichloromethane (DCM) were dried over 3 Å molecular sieves before use. Chromatography was performed using 60 Å silica gel. Compound **13** was prepared as described elsewhere[47]. Compounds **14, 16** and **18** were prepared following a previously described procedure[47,48]. ¹H and ¹³C NMR spectra were collected using at 500 and 125 MHz, respectively on a Bruker spectrophotometer. Spectra were

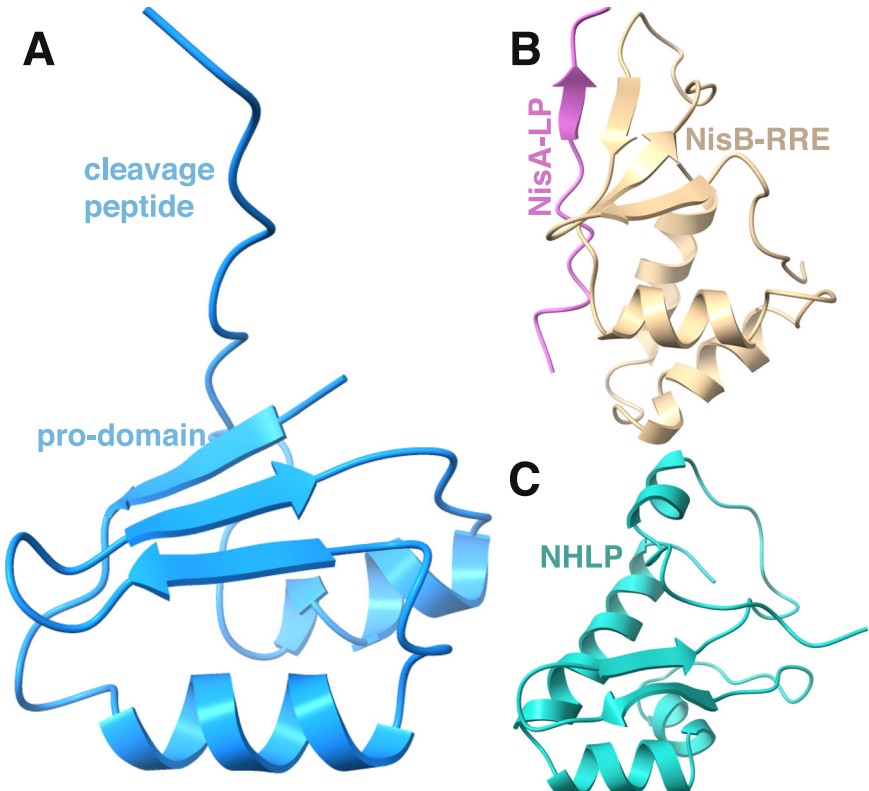

**Fig. 7 | Structural comparisons of substrate delivery domains.** Structural comparison between the pro-domain of CylA (**A**), the NisB-RRE bound to its substrate (**B**), and the NHLP-like LP found in the proteusins (**C**). Structural similarities highlight a convergent strategy for delivering a substrate to its cognate catalytic domain.

referenced to the residual chloroform solvent peak (7.26 ppm, $^1$H; 77.0 ppm, $^{13}$C). Matrix-assisted laser desorption/ionization time-of-flight mass spectrometry (MALDI-TOF MS) analysis was performed on a Bruker UltrafleXtreme MALDI-TOF mass spectrometer (Bruker Daltonics) at the University of Illinois School of Chemical Sciences Mass Spectrometry Facility. Fluorescence intensities and absorbances were measured using a Biotek Synergy H4. Circular dichroism data was collected using a Jasco J-1500 spectrophotometer. mCylL$_L$ and mCylL$_S$ were prepared as described previously from publicly available plasmids (Addgene #208759; #208760)[33]. Overnight cultures of *Enterococcus faecalis* were grown at 37 °C in brain heart infusion (BHI) media from colonies grown on BHI agar overnight at 37 °C. *E. faecalis* FA2−2 and mutants thereof were kindly provided by the Gilmore lab. pRSFDuet-1/CylA-27-412 was prepared as decribed previously[44]. *Escherichia coli* was cultured in lysogeny broth (LB) prepared according to the Miller formula. Primers were purchased from IDT. Any biological materials that are not available commercially can be obtained from the authors upon request.

### High-resolution tandem mass spectrometry
High resolution LCMS analysis was performed using an Agilent 1290 LC-MS QToF instrument. LC conditions are describe in each experimental method section. For all experiments, mass spectra were collected in the positive mode and 10 spectra/s were collected at 100 ms/spectrum. Tandem MS-MS fragmentation was conducted using normalized collision energies of 20 and 30. Data analysis was performed using the MassHunter Qualitative Analysis software (v 10.0) provided by Agilent.

### *tert*-Butyl (*R*)-4-(((*R*)-*tert*-butylsulfinyl)amino)-4-(4,4,5,5-tetramethyl-1,3,2-dioxaborolan-2-yl)butanoate (15)
A flask was charged with tricyclohexylphosphine tetrafluoroborate, toluene (0.8 mL), aqueous CuSO$_4$ (1.22 mL of a 30 mM solution;

1.2 mol%) and benzylamine (16.8 µL, 0.152 mmol, 5.0 mol%). The contents were stirred at room temperature for 10 min before adding a solution of **13** (800 mg, 3.06 mmol, 1.0 equiv.) dissolved in toluene (7.2 mL). To this mixture, was added bis(pinacolato)diboron (1.55 g, 6.12 mmol, 2.0) and the resulting suspension was stirred at room temperature overnight. The reaction mixture was diluted in ethyl acetate (50 mL) and filtered through a plug of deactivated silica gel (100:35 silica:water) eluting with ethyl acetate. The filtrate was concentrated via rotorary evaporation and the residue was loaded onto a deactivated silica gel column pre-equilibriated with DCM. A gradient of 100% DCM to 50% DCM/50% ethyl acetate was used. The eluted **15** was detected using KMnO$_4$ stain. This procedure yielded **15** as a colorless oil (655 mg, 1.68 mmol, 55% yield). $^1$H NMR (300 MHz, CDCl$_3$) δ 3.36 (d, *J* = 6.8 Hz, 1H), 3.06 (q, *J* = 6.9 Hz, 1H), 2.39 (t, *J* = 7.8 Hz, 2H), 1.96 (m, 2H), 1.45 (s, 9H), 1.29 (m, 12H), 1.21 (s, 9H).$^{13}$C NMR (125 MHz, CDCl$_3$) δ 172.95, 84.34, 80.47, 56.37, 55.67, 33.05, 28.33, 25.24, 25.15, 24.76, 22.77.

### (*R*)-4-(*tert*-butoxy)-4-oxo-1-(4,4,5,5-tetramethyl-1,3,2-dioxaborolan-2-yl)butan-1-aminium chloride (17)
Compound **15** (65 mg, 0.167 mmol, 1 equiv.) was dissolved in 1,4-dioxane (0.83 mL) and to this solution was added methanol (69 µL, 1.67 mmol, 10 equiv.) followed by HCl (4.0 M solution in 1,4-dioxane, 67 µL, 0.267 mmol of HCl, 1.7 equiv.). The resulting mixture was stirred at room temperature for 1 h before dilution with toluene (1 mL) and concentrating to dryness. The crude residue was immediately used in the synthesis of **1-10**.

### General Fmoc solid-phase peptide synthesis conditions
Fmoc SPPS was performed using a Liberty Blue™ peptide synthesizer (CEM). All resins were purchased preloaded from Chem-Impex and loading was determined immediately before starting the synthesis. All steps were performed at room temperature. Fmoc removal was

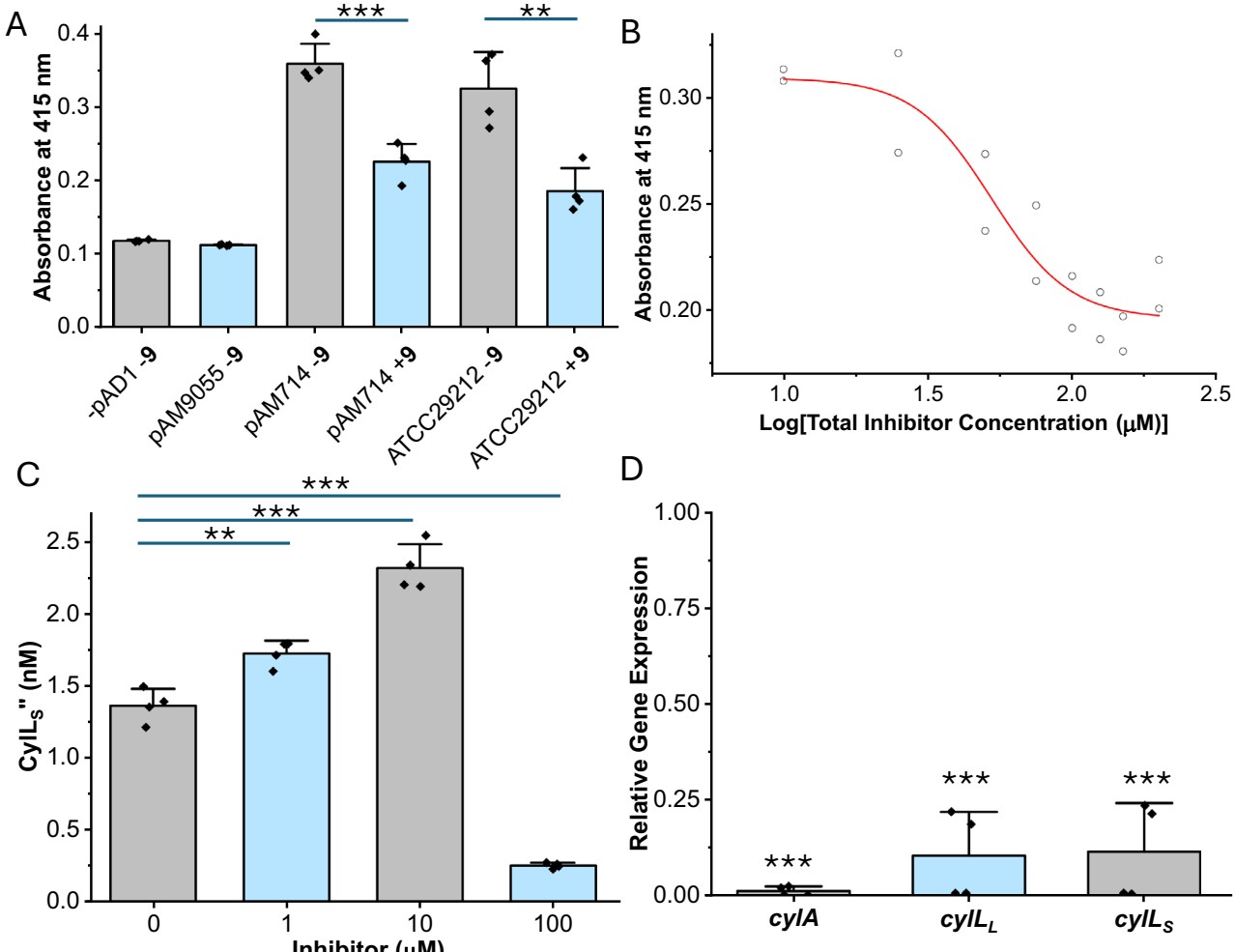

**Fig. 8 | Compound 9 inhibits the hemolytic activity of C+ *E. faecalis* by decreasing the amount of extractable CylL_S'' and reducing the level of cytolysin expression. A** Treatment of an inoculum ($OD_{600}$ = 0.35) of washed *E. faecalis* cells with 100 μM inhibitor **9** ( + **9**), in the presence of washed rabbit erythrocytes. The hemolytic activity decreased relative to untreated controls (− **9**). In total, four biological replicates (*n* = 4) for each condition were performed. Error bars represent the standard deviation and are centered at the mean. **B** Assessment of the ability of inhibitor **9** to block the hemolytic activity of live *E. faecalis* over a range of inhibitor concentrations under conditions described for (**A**). $EC_{50}$ was determined by fitting the data from two biological replicates (*n* = 2) with the dose-response function using Origin Pro 2024. Errors in the $EC_{50}$ are the errors calculated during the non-linear concatenate fit. **C** LCMS analysis of the concentration of CylL_S''

extracted from overnight cultures of C+ *E. faecalis* ATCC 29212 treated with **9** or vehicle. CylL_S'' extract concentrations were determined by comparison to an internal standard of $^{15}$N,$^{13}$C-labeled CylL_S''. Four biological replicates were performed (*n* = 4). Errors bars are centered at the mean and represent the standard of deviation. **D** RT-qPCR data showing the amounts of *cylL_S*, *cylL_L*, and *cylA* transcripts extracted from overnight cultures of C+ *E. faecalis* treated with 100 μM of compound **9** relative to an untreated control (relative gene expression of 1.0). Four biological replicates were performed (*n* = 4). Error bars are centered at the mean and represent the standard of deviation. Statistical significance was calculated using a two sample and two sided *t* test for (**A**) and (**C**) and a one sample one sided *t* test for (**D**). \*\**P* < 0.01, \*\*\**P* < 0.001. Source data are provided as a Source Data file.

performed by treating resin bound Fmoc protected peptide with 20% piperidine/DMF (3 ×10 min) followed by washing the resin with DMF (6 ×1 min). Fmoc-protected amino acids were coupled to the solid phase by mixing the resin with a solution containing Fmoc protected amino acid (3 equiv.), ethyl cyanohydroxyiminoacetate (3 equiv.) and *N,N′*-diisopropylcarbodiimide (DIC) (3 equiv.) in DMF for 1 h before draining and washing the resin with DMF (6 × 1 min). This process was repeated until the final amino acid was coupled. When desired, N-terminal acetylation was perfomed by removing the Fmoc group from the N-terminus then treating the resin bound peptide with resin capping solution (90:8:2 DMF:acetic anhydride:DIPEA) for 10 min. The resin was washed with DMF (6 × 1 min) then DCM (6 × 1 min).

### Synthesis of 1-10 starting from 1a-9a

Resin bound precursor peptides **1a**-**9a** were treated with 1% TFA/DCM for 20 min to cleave the peptide from the solid support. The resin was

removed by filtration and washed with additional 1% TFA/DCM. Solvent was removed via rotorary evaporation and the residue was dissolved in 1:1 DCM:toluene and concentrated again to remove trace TFA. For the synthesis of **1b, 2b** and **4b**-**9b**, DIPEA (3.0 equiv.) was added to solutions of freshly prepared **17** (1.3 equiv.) in dry DCM (final concentration of **1a**-**9a** was 0.05 M) and this solution was transferred to a 1 dram vial containing **1a, 2a** or **4a**-**9a** and HATU (1.2 equiv.) was added. For the synthesis of **3b**, the same reactions were setup using the same conditions but **18** was used in place of **17**. In all cases, the formed suspension was stirred for 13 h at room temperature, then diluted 10-fold with DCM and washed an equal volume of water. The organic layer was dried over sodium sulfate and concentrated. The crude peptide residue was dissolved in cleavage cocktail (TFA:triisopropylsilane:H$_2$O 95:2.5:2.5) to a concentration of 18 mg of crude peptide per milliliter. The solution was stirred for 1 h at room temperature then concentrated with a stream of N$_2$. Peptide was precipitated from the

concentrated residue through the addition of ice cold diethyl ether and the precipitate was pelleted by centrifugation (4000 g for 10 min) then washed with another volume of diethyl ether. The peptide residue was dissolved in water and loaded onto a Bond Elut C18 Solid Phase Extraction (SPE) Cartridge. The stationary phase was washed with $H_2O$ + 0.1% TFA then the peptide was eluted with 60% $CH_3CN$ + 0.1% TFA/40% $H_2O$ + 0.1% TFA. The elution fraction was lyophilized and the lyophilized peptide was dissolved in water prior to purification.

## Purification of peptides 1−10

Crude peptides **1-10** were purified by reversed-phase UHPLC using a ThermoFisher Vanquish UHPLC using an Accucore™ C18, 2.6 μm, 150 ×4.6 mm column and a flow rate of 0.4 mL/min. Solvent A was $H_2O$ + 0.1% trifluoroacetic acid (TFA) and solvent B was $CH_3CN$ + 0.1% TFA. The same gradient method was used to purify each peptide: 2% solvent B for 2 min followed by a linear ramp to 20% solvent B over 8 min. Fractions containing pure peptide were determined by LCMS and combined and concentrated by lyophilization. Lyophilized peptide was dissolved in $H_2O$ and characterized by high-resolution LCMS (Supplementary Figs. 1-2). A Poroshell C18 2.7 μm 120 Å 100 × 3.0 mm column was used with the following method: 98% $H_2O$ + 0.1% FA (solvent A)/2% $CH_3CN$ + 0.1% FA (solvent B) for 2 min then a linear gradient from 98:2 to 2:98 solvent A:solvent B over 6 min. The retention times of **1-10** are included in Supplementary Table 2. The stock concentration of **1** was determined gravimetrically, the concentrations of stocks of **2-10** were determined by comparing the absorbance of each sample at 205 nm to the absorbance of **1**. Absorbances at 205 nm were determined from the peak areas of traces collected using the UV-detector equipped to the LCMS. The isolated yields of peptides **1, 3-9** were 54%, 67%, 75%, 81%, 62%, 51%, 77%, 65%, and 63%, respectively. For peptide **2**, additional purification was necessary.

After UHPLC, crude **2** (20 mg) was dissolved in dry DMF (2 mL) and incubated with 1-glycerol resin (200 mg, 1.1 mmol/g) that was pre-swelled in DMF. The mixture was incubated for 13 h at room temperature then the resin was filtered off and washed with dry DMF (6 × 1 min). The resin was treated with 2 mL of elution cocktail (DCM:methanol:$H_2O$ = 5:4:1) for 3 ×30 min and the combined elutions were concentrated via rotary evaporation. The crude residue was dissolved in $H_2O$ and lyophilized. The lyophilized peptide **2** was characterized by high-resolution LCMS (Supplementary Table 1). Stock concentrations were measured by LCMS through comparison to a standard using absorbance at 205 nm. Isolated yield of **9** was 19% based on resin loading.

UPLC traces of purified **1-10** are provided in Supplementary Fig. 20 and the Source Data file.

## Synthesis and purification of the FRET substrate peptide

The synthesis used Fmoc-L-Gly-Wang loaded polystyrene resin, which was extended to the penultimate residue (Gly) using standard Fmoc SPPS conditions. Fmoc-L-Asp(EDANS)-OH was coupled to the N-terminus by incubating the resin bound peptide with a solution of Fmoc-L-Asp(EDANS)-OH (3 equiv.), benzotriazolyloxy-tris[pyrroli-dino]-phosphonium hexafluorophosphate (3 equiv.) and DIPEA (3 equiv.) dissolved in dry DMF at room temperature for 13 h. The resin was washed with DMF (6 × 1 min) then DCM (6 × 1 min), and the peptide was cleaved from the resin and globally deprotected through treatment with cleavage cocktail (TFA:phenol:$H_2O$ 90:5:5) for 90 min at room temperature. TFA was removed under a stream of $N_2$ and the crude peptide was precipitated from this residue through the addition of ice-cold diethyl ether. The precipitated peptide was pelleted by centrifugation (4000 g for 10 min) then washed with another volume of cold diethyl ether. The pelleted peptide was dissolved in 1:1 $H_2O$:$CH_3CN$ and purified by semi-preparative R-HPLC using a Macherey-Nagel NUCLEODUR C18 HTec, 10 μm, 250 ×10 mm column. Solvent A was $H_2O$ + 0.1% TFA and solvent B was $CH_3CN$ + 0.1%TFA. A

gradient method was used which started with 2% solvent B for 10 min followed by a linear ramp from 2% to 80% solvent B over 25 min and then another linear ramp from 80% to 100% solvent B. Fractions were analyzed by MALDI-TOF MS and those containing pure peptide were combined and lyophilized. Lyophilized peptide was dissolved in 1:1 $CH_3CN$:$H_2O$ and characterized by LCMS-MS (Supplementary Figs. 4 and 5A). FRET peptide concentration was determined by the absorbance of DABCYL at 482 nm using an extinction coefficient of 25500 cm$^{-1}$ M$^{-1}$[68]. Performing this synthesis on a 0.1 mmol scale yielded 47 mg of the FRET peptide (26% yield).

## Expression and purification of His$_6$-CylA-27-412

*Escherichia coli* BL21 (DE3) cells were transformed with pRSFDuet-1/CylA-27-412 and plated on LB containing 50 mg/L kanamycin. A single colony was picked and used to inoculate 20 mL of LB containing 50 mg/L kanamycin which was incubated overnight at 37 °C. The overnight culture was used to inoculate 1 L of LB (1:100) which was incubated at 37 °C until an $OD_{600}$ of 0.8 was reached. The culture was then cooled on ice for 30 min and IPTG was added to a final concentration of 250 μM. The flask was incubated at 18 °C for 16 h before collecting the cells by centrifugation (6000 g for 15 min). The cell pellet was resuspended in CylA lysis buffer (20 mM HEPES, 1 M NaCl, 10% glycerol, pH = 7.5) and the cells were lysed by sonication (8 rounds of the 30 s on followed by 30 s off). The cell lysate was centrifuged (120,000 g for 45 min) and the pellet was resuspended in pellet extraction buffer (8 M urea, 1 M NaCl, 20 mM HEPES, 10% glycerol, pH = 7.5). The pellet suspension was sonicated (7 rounds of the 30 s on followed by 30 s off) then centrifuged (50,000 g for 45 min). The supernatant was collected and dialysed against Dialysis buffer (20 mM HEPES, 300 mM NaCl, pH = 7.5) overnight. Percipitate was removed by centrifugation (120,000 g for 45 min) and the supernatant was subjected to immobilized metal affinity chromatography (IMAC) using Ni-resin as described previously[44]. Elution fractions containing His$_6$-CylA-27-412 were combined and dialyzed against Dialysis buffer. The IMAC purified protein was incubated at 4 °C until SDS-PAGE showed that the sample underwent autolysis (usually 48 h). His$_6$-CylA-27-412 was further purified by size exclusion chromatography (SEC) using a Super-dex 75 column (Cytiva). Fractions containing His$_6$-CylA-27-412 were concentrated via 3-kDa MWCO centrifugal filter.

## Crystallization and structure determination of His$_6$-CylA-27-412

The concentrated protein from the size exlusion chromatography was used for crystallization by the sitting drop vapor diffusion method. His$_6$-CylA-27-412 purified by size exclusion chromatography was concentrated to 15.4 mg/ml and mixed in 1:1 ratio (v/v) with a solution of 0.2 M ammonium sulfate, 0.1 M sodium acetate trihydrate pH 4.2, 22% (w/v) polyethylene glycol 2000 monomethyl ether and incubated against the latter solution using the sitting drop vapor diffusion method. Large rhombohedrally shaped crystals were obtained after 4 days and were subsequented transferred to the crystallization solution supplemented to 30% (w/v) polyethylene glycol 2000 monomethyl ether before vitrification by direct immersion into liquid nitrogen.

X-ray diffraction (XRD) data were collected at the Cornell High Energy Synchrotron Source (Beam line ID7B2 MacCHESS, Ithaca, NY) using a Dectris Pilatus 6 M detector system. Diffraction data were integrated and scaled using the XDS package[69] as implemented in autoProc[70]. The diffraction data shows slight anisotropy below the 1.35 Å Bragg cutoff owing to modest completion in the highest resolution shell. The high resolution data were included as the data contributed to the quality of the resultant electron density maps. Crystallographic phases were determined by the molecular replacement method as implemented in Phaser[71] using an AlphaFold[60] model on the catalytic domain of CylA. Automated using wARP[72] and manual model building using COOT[73], interspersed with rounds of crystallographic refinement using REFMAC5[74] resulted in the final model.

Cross-validation, using 5% of the data for the calculation of the free R factor, was utilized throughout model building process in order to monitor building bias. The stereochemistry of all of the models was routinely monitored throughout the course of refinement using PROCHECK[75]. Relevant data collection, phasing, and refinement statistics are provided in Supplementary Table 4. The structure has been deposited at the Protein Databank as PDB ID: 9EKQ.

## Analytical size exclusion chromatography

Samples of His$_6$-CylA-27-412 prepared as described in 'Expression and purification of His$_6$-CylA-27-412' were diluted to a concentration of 50 µM and injected (20 µL injection volume) into an Agilent 1260 infinity analytical high-performance liquid chromatography system equipped with a Biozen™ 1.8 µm, 300 × 4.6 mm LC column pre-equilibriated with aSEC buffer (100 mM NaHPO$_4$, 200 mM NaCl, pH = 7.4). Absorbance at 220 nm was monitored. A flow rate of 0.25 mL/min was used for all experiments. Retention time was correlated to protein size using a Gel Filtration Standard (Biorad) and samples of carbonic anhydrase isolated from bovine erythrocytes (Sigma Aldrich) and albumin isolated from bovine serum (Sigma Aldrich).

## Analysis of CylA inhibition by LCMS

His$_6$-CylA-27-412 was added to PBS (pH = 7.4) to achieve a total enzyme concentration of 4 nM then **1** was added to a concentration 1 µM. The mixture was incubated at room temperature for 10 min before adding biosynthetically prepared mCylL$_L$ or mCylL$_S$ to a concentration of 10 µM. Reactions were allowed to stand for 30 min before quenching through adjustment to 1% formic acid. Reactions were analyzed directly by LCMS using a Jupiter 5 µm C4 300 Å, 50 × 2 mm LC column. The following gradient was used: 98% H$_2$O + 0.1% FA (solvent A)/2% MeCN + 0.1% FA (solvent B) for 2 min then 98:2 to 2:98 solvent A:solvent B over 10 min. Extracted ion chromatograms (EIC) for each species were merged into a single trace. In the assays without inhibitor, mCylL$_L$ was a slightly better substrate than mCylL$_S$, as previously reported[44], suggesting the enzyme has a preference for the sequence of the former. CylA was also previously reported to prefer the modified peptides over the linear peptides[44].

## Co-operative pore formation assay

Pyranine encapsulated liposomes were prepared as described elsewhere[21]. All fluorescence measurements were collected using a BioTek Synergy H4 plate reader. mCylL$_L$ and mCylL$_S$ were prepared as described elsewhere[44]. To a well in a black 96-well plate (200 µL) was added 90 µL high pH buffer (5 mM MES, 5 mM Tricine, 5 mM NaCl, pH 7.4) containing CylA (4 nM final concentration). After adding inhibitor, the mixture was incubated at room temperature for 10 min then freshly prepared (no more than 2 days old) pyranine encapsulated DOPC liposomes were added such that the total DOPC concentration was 80 µM. The pyranine fluorescence signal (excitation wavelength = 460 nm, emission wavelength = 510 nm) was monitored for 8 min at room temperature then a 1:1 mixture of mCylL$_S$:mCylL$_L$ was added such that the total peptide concentration was 500 nM. Pyranine fluorescence was monitored for 30 min then the Triton X-100 concentration was adjusted to 0.1% and the fluorescence intensity was monitored for another 5 min. The resulting curves were max-min normalized yielding the fraction of permeabilized liposomes. The maximum fraction permeabilized achieved during the 30 min period in between adding cytolysin and adding Triton X-100 was used as a measure of enzyme activity. This experiment was performed in duplicate or triplicate for each inhibitor at each concentration.

## Determination of the K$_m$ for the FRET peptide

To determine its K$_m$, FRET peptide (125 µM to 2 µM) was combined with His$_6$-CylA-27-412 (20 nM) in PBS (pH = 7.4) at 37 °C and the EDANS fluorescence signal (excitation wavelength = 336 nm, emission

wavelength = 490 nm) was monitored for 4 h using a BioTek Synergy H4 plate reader. The slope of the linear region of each curve was used as a metric of initial velocity and the data was fit to the Hill form of the Michaelis-Menten model yielding K$_m$ and the Hill coefficient (h).

## Determining initial rates using the FRET peptide substrate

His$_6$-CylA-27-412 (20 nM) in PBS (pH = 7.4) was combined with inhibitor or mCylL$_L$, mCylL$_S$, or a stoichiometric mixture of mCylL$_L$:mCylL$_S$ at various concentrations in a 96 well plate and the plate was incubated at 25 °C for 10 min before adding the FRET peptide substrate (10 µM). The emission intensity of EDANS was measured over a 4 h period using a BioTek Synergy H4 plate reader. The slope of the linear portion of the emission intensity as a function of time curve was calculated ($v_i$), divided by the slope of an uninhibited control ($v_o$), and the ratio ($v_i/v_o$) was plotted as function of total inhibitor concentration.

## Activity assays with C+ *Enterococcus faecalis*

Rabbit erythrocytes (Hemostat) were washed with cold, sterile PBS and diluted in BHI to a concentration of 1% v/v. A 3 mL overnight culture of *E. faecalis* was centrifuged (4000 × g, 10 min) and the pellet was resuspended in 7 mL of BHI and the suspension was centrifuged (4000 × g, 10 min). The pellet was resuspended in BHI to achieve an OD$_{600}$ of 1.4 and 50 µL of this suspension was added to a well containing 50 µL of inhibitor (400 µM). The inhibitor and the *E. faecalis* were incubated at room temperature for 10 min before adding 100 µL of 1% v/v rabbit erythrocytes in BHI. The inhibitor-free control well contained the same concentration of bacteria and rabbit erythrocytes. The sterile negative control and sterile positive control contained 100 µL of BHI and 100 µL of 1% v/v rabbit erythrocytes. The plate was incubated at 37 °C without shaking until complete hemolysis was observed in the inhibitor free wells (usually 4 h). Triton X-100 was then added to the sterile positive control wells to a concentration of 0.03% and the plate was incubated for another 10 min. The plate was then placed on ice and centrifuged at 1000 × g for 10 min to pellet all the remaining cellular material. The supernatant was diluted 10-fold in PBS and the absorbance at 415 nm was measured. For each condition, four wells were prepared and analyzed. To determine an EC$_{50}$ the same experiment was performed with the same controls but inhibitor concentrations equal to 200, 150, 125, 100, 75, 50, 25, and 10 µM were used.

## Growth curve assay

To a 96-well plate, BHI media was added containing 200 µM of inhibitor **9**. An overnight culture of *E. faecalis* was diluted to a concentration of 10$^6$ CFU/mL and an equal volume of this culture was added to the well containing inhibitor. A growth control was prepared by combining the diluted bacterial cells with an equal volume of sterile BHI. A sterility control well was filled with sterile BHI to the same volume. The 96-well plate was transferred to a plate reader preheated to 37 °C and the plate was shaken continuously. The OD$_{600}$ of each well was measured over the course of 24 h. Uncorrected OD$_{600}$ values were plotted as a function of time.

## Cytoxicity assays with HeLa cells

Cell culturing and maintenance was performed in Dulbecco's Modified Eagle Medium (DMEM, Gibco) supplemented with 10% heat inactivated fetal bovine serum (FBS, Gibco). Cells were routinely checked for mycoplasm using mycoplasm specific primers. HeLa cells (ATCC) were seeded in 96 well plates (Nunc) using a seeding density of 5000 cells/well and allowed to adhere for 16 h at 37 °C, 5% CO$_2$. The next day, cells were treated with compounds **2, 3, 5-9** in 2-fold dilutions, starting at 100 µM. Post-treatment, plates were incubated for 72 h at 37 °C, 5% CO$_2$. Subsequently, the growth medium was removed, and the cells were washed once with pre-warmed phosphate buffered saline (PBS; pH = 7.2). Next, 50 µL of DMEM-FBS media containing 1 mg/mL of 3-(4,5-dimethylthiazol-2-yl)-2,5-diphenyltetrazolium bromide (MTT) was

added to the cells and the cells were incubated in the dark at 37 °C, 5% $CO_2$ for 4 h. During this time, live cells reduced yellow tetrazolium salts to purple formazan crystals through NAD(P)H-dependent oxidor-eductase activity. After 4 h, the medium was removed carefully and the insoluble formazan crystals were solubilized by incubating with 100 μL of dimethyl sulfoxide (DMSO) in the dark for 1 h with intermittent swirling. Finally, absorbance was recorded at 575 nm (Supplementary Fig. 18). The percentage of cells that survived over the course of the assay was calculated using Eq. 1.

$$\%survival = 100\% - \frac{(Absorbance\ of\ treated\ well - Avg(Absorbance\ of\ negative\ control\ wells))}{(Avg(Absorbance\ of\ positive\ control) - Avg(Absorbance\ of\ negative\ control))} * 100\% \quad (1)$$

### Detection of CylL$_S$" from extracts of *E. faecalis*

An overnight culture of *E. faecalis* ATCC 29212 was diluted to an $OD_{600}$ equal to 2.0 in BHI then further diluted 1000-fold in BHI. Inhibitor **9** or vehicle was added immediately after dilution and the inoculated samples were incubated with shaking at 37 °C for 20 h. Following incubation, a 400 μL aliquot of each sample was flash frozen and lyophilized. Each lyophilized sample was extracted with 400 μL of 90% methanol/10% water containing $^{15}N,^{13}C$-labeled CylL$_S$" (5.8 nM). Alternating rounds of water bath sonication and vortexing were used to homogenize the samples. Once homogenized, the samples were allowed to stand at room temperature for 1 h then were centrifuged at 16,000 g for 10 min, incubated at −20 °C for 1 h, and centrifuged at 16,000 g for 10 min. The supernatant was analyzed by LCMS using injection volumes of 5 μL. EIC corresponding to the exact masses of $[M + 2H]^{2+}$ and $[M+Na+H]^{2+}$ were integrated and the peak areas were added together giving the total CylL$_S$" and internal standard responses. The concentration of CylL$_S$" was estimated from the response ratio multiplied by the concentration of the internal standard.

### Assessing inhibitor stability

Inhibitor **9** (100 μM) was incubated in RPMI 1640 Medium (Gibco) containing 25% human serum (Sigma-Aldrich) for 3 h at 37 °C then flash frozen and lyophilized. Similarly, **9** was diluted to 100 μM in BHI that was inoculated with *E. faecalis* ATCC 29212 before incubating at 37 °C for 20 h. The overnight culture was flash frozen and lyophilized. The lyophilized samples were extracted with a volume of 90% methanol/10% water that was equal to the volume of the sample prior to lyophilization. The extract was assumed to have a concentration of 100 μM. The inhibitory activity of the extract was determined using the procedure described in the 'Determining initial rates using the FRET peptide substrate' section of the materials and methods.

### Determining the relative amounts of *cylL$_L$*, *cylL$_S$*, and *cylA* in total RNA extracts from *E. faecalis*

The amounts of *cylL$_L$*, *cylL$_S$*, and *cylA* transcripts relative to the amount of 16S rRNA were determined by RT-qPCR. Amplicons and primers used to quantify *cylL$_L$*, *cylL$_S$* and 16S rRNA were identical to those previously reported[26]. All primers are listed in Supplementary Table 3. Overnight cultures of *E. faecalis* ATCC 29212 were prepared as described in 'Detection of CylL$_S$" from extracts of *E. faecalis*'. Upon inoculation, samples were treated with 100 μM of **9** or vehicle. The samples were incubated at 37 °C for 20 h, then cells were collected by centrifugation (5000 g for 10 min). The supernatant was removed and the cells pellets were re-suspended in Lysis buffer (10 mM TrisHCl, 3 mg/mL lysozyme, pH = 7.8) at a concentration of 160 μL of Lysis buffer for 1 mL of over-night culture. The cells were incubated in Lysis buffer at room temperature for 15 min then RNA was extracted using a Monarch® Total RNA Miniprep Kit. Isolated RNA was immediately converted to cDNA using an iScript™ cDNA Synthesis Kit. cDNA was amplified using a SsoAdvanced Universal SYBR Green Supermix and a StepOnePlus Real-

Time PCR System (Applied Biosystems). The expression level of each gene in each sample was determined by taking the difference between the $C_T$ of the amplification curve for the gene of interest (*cylL$_L$*, *cylL$_S$* or *cylA*) and the $C_T$ of the amplification curve for 16S rRNA, yielding $\Delta C_T$. $\Delta\Delta C_T$ was determined by taking the difference between the $\Delta C_T$ of the inhibitor treated sample and the $\Delta C_T$ of the inhibitor-free control for each gene. Fold change in gene expression was calculature using Eq. (2)[76]. Four biological replicates were performed.

$$Fold\ change\ in\ gene\ expression = 2^{\Delta\Delta CT} \quad (2)$$

### Reporting summary

Further information on research design is available in the Nature Portfolio Reporting Summary linked to this article.

### Data availability

The data generated in this study have been deposited in the Mendeley database[77] (https://doi.org/10.17632/yh5kp2fzjn.1) and are also included in a Source Data file provided with this study. The source data file contain data used to prepare Table 1, Supplementary Table 1, Figs. 3, 5, 8 and Supplementary Figs. 1-7, 9-10, 17-29. Atomic coordinates have been deposited in the Protein Data Bank (PDB) under accession codes PDB 9EKQ. Atomic coordinates of subtilisin BPN', NisB, LicP, NisP, and EpiP discussed in this study are available under accession codes 1SPB, 4WD9, 4ZOQ, 4MZD, 3QFH, and 8TB1, respectively. Source data are provided with this paper.

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

## Acknowledgements

This study was supported by the Howard Hughes Medical Institute (W.A.v.d.D.), and the National Institute of General Medical Sciences (R35 GM151874 to S.K.N.). A Bruker UltrafleXtreme MALDI TOF/TOF mass spectrometer was purchased in part with a grant from the National Center for Research Resources, National Institutes of Health (S10 RR027109 A). W.A.v.d.D. is an Investigator of the Howard Hughes Medical Institute (HHMI). R.M. was supported by a postdoctoral fellowship from the Life Sciences Research Foundation and a postdoctoral fellowship from the Natural Sciences and Engineering Research Council of Canada. Y.Y. was supported by a Spudich Undergraduate Research Scholarship in Chemistry. This study is subject to HHMI's Open Access to Publications policy. HHMI laboratory heads have previously granted a nonexclusive CC BY 4.0 license to the public and a sublicensable license to HHMI in their research articles. Pursuant to those licenses, the author-accepted manuscript of this article can be made freely available under a CC BY 4.0 license immediately upon publication.

## Author contributions

R.M. designed the study, selected the target, designed and synthesized the inhibitors, conducted inhibition assays, and wrote the first draft of the manuscript. B.C. expressed, purified and crystallized enzyme for XRD. Y.Y. assisted with the synthesis of inhibitors. C.P. performed toxicity assays against HeLa cells and helped develop the inhibitor purification method. S.K.N. analyzed XRD data, solved the structure of CylA and contributed to writing the maniscript. W.A.V. motivated the study, provided guidance as the work developed and contributed to writing the manuscript. M.S.G. provided mutants of *E. faecalis* for bioactivity studies. All authors contributed to the preparation of the manuscript.

## Competing interests

The University of Illinois has filed a provisional patent application on the work described in this study: van der Donk, W.A.; Moreira, R.; Chakraborty, B.; Nair, S.K.; "INHIBITION OF GUT BACTERIA" U.S. Patent Application No. 63/779,530. Filed March 28, 2025. This application covers CylA as target for therapeutic intervention.
