## [Transparent Peer Review file · Nature Communications]

Combating Virulent Gut Bacteria by Inhibiting the Biosynthesis of a Two-component Lanthipeptide Toxin

Corresponding Author: Professor Wilfred van der Donk

Version 0:

Reviewer comments:

Reviewer #1

(Remarks to the Author)

This study by Ryan Moreira et al. focuses on enterococcal cytolysin, a toxic peptide produced by *Enterococcus faecalis*, a bacterium commonly found in the human gut. The researchers aimed to develop small-molecule inhibitors to prevent cytolysin activation, thereby reducing its harmful effects. They targeted CylA, a key enzyme responsible for activating the toxin. By designing and synthesizing a series of inhibitors, they demonstrated that these compounds, particularly compound 9, could block cytolysin maturation at low concentrations. Further experiments confirmed that the inhibitor effectively reduced cytolysin's harmful activity without affecting bacterial growth. These findings highlight the potential of targeting CylA as a strategy to mitigate diseases caused by cytolysin-producing *E. faecalis*. Overall, the study is well-designed, comprehensive, and convincing.

Major Points:

1. The writing is quite technical, which may make it challenging for non-specialists or those outside the field of peptide chemistry to follow. While the technical level is appropriate for a scientific manuscript, some sentences are overly complex and could be simplified. Breaking these into shorter, clearer statements would improve overall readability. Additionally, providing a more general description of the outcomes would make the study's interesting results more accessible to a wider audience.

2. Line 417. The hypothesis of a dynamic substrate recognition site is intriguing, but no experimental evidence (e.g., mutagenesis, binding assays, or NMR data) has been provided to confirm structural changes in solution. Since no inhibitor-bound structure was obtained, an alternative approach, such as molecular dynamics simulations, could be suggested to support the hypothesis.

3. How can the authors rule out that factors other than cytolysin contribute to erythrocyte lysis in *E. faecalis*? It is possible that compound 9 also inhibits other, yet unknown, (virulence) factors as a side effect. To confirm that the observed lysis effects on erythrocytes are specifically due to cytolysin in the bacterial supernatants, including a cytolysin mutant as a control is essential. This mutant control is preferable to simply omitting bacteria from wells containing erythrocytes.

Minor Points:

1. Line 178. The explanation that up to 30% of the deprotected peptide underwent protodeboronation is clear. However, it would be helpful to briefly discuss any implications this may have for the stability of the inhibitor.

2. Why was Fmoc-SPPS chosen over other methods, such as Boc/OBzl?

3. Line 236. The finding that compound 9 (cyclopropylglycine variant) was more potent than the parent inhibitor is interesting but is not deeply explored. Could this be due to altered conformational flexibility or enhanced binding interactions?

4. Consistency: warhead or war head

Reviewer #2

(Remarks to the Author)

Cytolysin is peptide produced by G+ *Enterococcus faecalis*. This *E. faecalis* strain is associated with significant mortality in patients with alcohol-induced hepatitis. A long and a short isoform of cytolysin is initially synthesized by the ribosome followed by subsequent posttranslational modifications involving sequential processing by the enzymes CylM, CylB and CylA. CylA is a secreted serine protease which cleaves a hexapeptide secretion signal with the sequence GDVQAE from both the long and short isoforms, thus producing the active cytolysins. The manuscript by Moreira et al. describes the development of boronic acid containing peptide inhibitors targeting CylA, which are derived from the hexapeptide leader sequence of the long and short cytolysin isoforms. The best peptide in which the valine at position 3 is replaced with a cyclopropylglycine exhibits an EC(50) values of ~10 nM, an ~3-fold reduction compared to the starting sequence. With selected inhibitors the authors performed detailed inhibition studies with the purified enzyme and also with C+ *E. faecalis* cells. Furthermore, the authors determined a high resolution crystal structure of the enzyme in which its pro-domain is still bound to the catalytic domain via its C-terminal tail. As the sequence of the C-terminal tail (PDFKPE) is related to the sequence of the peptide-based inhibitors, this structure provides insights how the inhibitors could be bound.

Overall, this is a very interesting study employing a variety of experimental techniques, which yielded high affinity starting compounds targeting an potential drug target. Listed below are points, which the authors should address in a potential revised version.

1. Why is the intrinsic activity of the enzyme to cleave especially mCylL(L) but also mCylL(S) so small (Fig. 3). Does the conversion achieve 100% after longer incubation times or incubation at higher temperatures? Why is there even such a pronounced difference between the long and short forms? This is rather surprising in light of the conservation of the two cytolysins around the cleavage site.
2. In the present form Fig. 4 is helpful to illustrate how the assay works, however, what one would really like to see instead would be the actual data. Please modify the figure so that the existing Fig. 4 becomes panel (A) of a revised Fig. 4 and in subsequent panels show the experimental data as presented in the current Fig. S3 for a subset of inhibitors. Data for all remaining inhibitors should be shown in a revised version of Fig. S3.
3. The aSEC data shown in Fig. 4D are rather puzzling, especially the shift of the two peaks. Assuming that the peak at ~11.4 min shifts to the right because the complex increases in size due to inhibitor binding, why is there a complete shift already at substoichiometric inhibitor concentrations? If it is not inhibitor binding, what causes this shift? For the peak at 11.8 min, why is there a shift to later elution times? Please provide explanations for these effects. Ideally, the aSEC data should be replaced with SEC-MALLS data if the authors can gain access to such an instrument. At the least, the authors should show the full chromatograms (maybe in the supplement) and also analyze the corresponding fractions on SDS-PAGE gels.
4. There are a few points regarding the structural studies. (a) Although not stated in the legend to Fig. 6B, this is probably a 2Fo-Fc map, which should be replaced with an omit map. (B) Please also show Ser359 in panel B to make it easier for the reader to see where the peptide is relative to the active site cysteine. (C) Based on the experimental structure please provide a model with the bound inhibitor so that the reader can judge the position of the boronic acid with respect to the active site serine. (D) Please describe and show (in a separate panel in Fig. 6) the interactions between the pro-domain and catalytic domain. What is the amount of buried surface in the interface? (E) Please replace all ColabFold predictions with AlphaFold3 predictions, as this represents the current state of the art. (F) I find Fig. 7 not very helpful to convey the similarities in the pro-domains. Please show a superposition of the three structures (probably in what in Pymol is called a ribbon representation). If needed, the cartoon representations could be shown at the same scale next to the superimposition. (G) Given the high resolution the R-factors of the refined model are somewhat disappointing. Did the authors try to refine individual anisotropic B-factors? (H) The authors should comment on the low completeness of the data in the highest resolution shell, which is probably caused by the detector distance being too large. Given the data collection statistics (R(pim), I/sig/ and CC(1/2)) the data extended apparently to even higher resolution. The authors need to comment on these aspects. (I) The section in Materials and Method should probably be entitled "Crystal structure analysis of His(6)-CaIA-27-412" and not "Crystallization and Data Collection of ..." since it also covers the structure solution, model building and refinement steps.
5. Please provide a better explanation for the data presented in Fig. 8C. Shouldn't the concentration of CylL(S) decrease in the presence of inhibitor and given the data presented in Fig. 8B this should occur for sure at a concentration of 10 µM? This unexpected behavior is referred to in the text as "bidirectional modulation" (line 500), which seems rather vague.

Additional points:

Line 43: Replace "colon-rectal" with "colorectal"

Line 731: Please indicate the instrument used for these experiments.

Fig. S1 and S2: Please add calculated masses of the inhibitors next to the chemical structures. Although the values are listed in Table S1, it will make it easier for the reader if the info is also included in the figure.

Table S3:

Given the high resolution it seems justified to state the unit cell dimensions with two digits after the decimal point.

Fig. S6: Please explain "vi" and "vo" and state that in the absence of inhibitor "vi/vo" is equal to one. Why are error bars

sometimes missing?

Fig. S7: The same reservations described for Fig. 5D apply here, i.e. a clear assignment of the peaks is missing. In addition, the whole chromatogram should be shown.

Fig. S9: Repeat prediction with Alphafold3 plus add prediction of crystal structure to get an idea about the accuracy of the prediction when compared to the experimentally derived structure.

Fig. S11: Where is the curve for the negative control?

Fig. S12 and S13: The legends are missing.

Reviewer #3

(Remarks to the Author)

This is an excellent study. The authors are addressing a problem of key relevance to chemical biologists and microbiologists seeking to combat cytolyisin virulence and develop treatments for cytolyisin-induced disease states. Based on the detailed studies of the biosynthesis of the two-component lantibiotic cytolyisin previously published by this group, this work identifies the extracellular serine protease CylA as a suitable target for preventing maturation of both of the peptide components of cytolyisin. A series of peptide boronic acids were designed and synthesised based on the sequences recognised by CylA, enabling identification of nanomolar inhibitors. A thorough study of the inhibition of CylA by inhibitor 9 is presented. Importantly, the authors have succeeded in obtaining an X-ray crystal structure of CylA bound to the proteolytically excised pro-domain, which provided insights into the enzyme-inhibitor interactions. The authors have provided well-written and comprehensive experimental details and characterisation data in the Experimental and in the Supporting Information.

The manuscript is well written and will be of interest to readers of Nature Communications. I recommend publication, subject to the authors addressing some minor points and queries:

There is an ambiguity in Figure 2. The synthesis of peptides 1b, 3b - 9b are shown as starting from 2-CITrt resin pre-loaded with Fmoc-L-Ala. However in peptide 6 this residue is Abu (and therefore this was presumably the residue pre-loaded on the resin)

The characterisation of the peptides is good (Supplementary Figures 1 and 2). To aid reproducibility, could the HPLC retention times for the purified peptides be included?

The rationale for using mCylLs and mCylLL as substrates for CylA is not made clear. As the *in vivo* substrates for CylA are CylLL' and CylLS', would it not have been more relevant to use these?

In the discussion of Figure 5D, the authors mention that several overlapping peaks are observed by aSEC when His6-CylA-27-412 is analysed in the presence of inhibitor 9. Could native ESI mass spectrometry have been useful here to identify both complexation and folding/unfolding?

Version 1:

Reviewer comments:

Reviewer #1

(Remarks to the Author)

The authors have satisfactorily addressed all suggestions and provided comprehensive responses to all queries. I therefore recommend that the manuscript be accepted for publication.

Reviewer #2

(Remarks to the Author)

The authors have addressed the concerns raised during the initial review. I noted two minor points when looking at the manuscript:

1. Three extra letters "ack" appear at the start of the title, which presumably need to be removed.
2. The error bars in Fig. 4d are too small to be easily visible.

Reviewer #3

(Remarks to the Author)

Thank you for providing the revised manuscript and a comprehensive response to the reviewers comments. I confirm that all of my concerns and suggestions have been addressed by the authors.

RESPONSE TO REVIEWER COMMENTS

Reviewer #1 (Remarks to the Author):

This study by Ryan Moreira et al. focuses on enterococcal cytolysin, a toxic peptide produced by *Enterococcus faecalis*, a bacterium commonly found in the human gut. The researchers aimed to develop small-molecule inhibitors to prevent cytolysin activation, thereby reducing its harmful effects. They targeted CylA, a key enzyme responsible for activating the toxin. By designing and synthesizing a series of inhibitors, they demonstrated that these compounds, particularly compound 9, could block cytolysin maturation at low concentrations. Further experiments confirmed that the inhibitor effectively reduced cytolysin's harmful activity without affecting bacterial growth. These findings highlight the potential of targeting CylA as a strategy to mitigate diseases caused by cytolysin-producing *E. faecalis*. Overall, the study is well-designed, comprehensive, and convincing.

RESPONSE: Thank you for the positive overall evaluation. We have made the revisions indicated below in response to the reviewer suggestions.

Major Points:

1. The writing is quite technical, which may make it challenging for non-specialists or those outside the field of peptide chemistry to follow. While the technical level is appropriate for a scientific manuscript, some sentences are overly complex and could be simplified. Breaking these into shorter, clearer statements would improve overall readability. Additionally, providing a more general description of the outcomes would make the study's interesting results more accessible to a wider audience.

RESPONSE: We thank the reviewer for these valuable comments. In response, we reduced the number of complex sentences throughout the manuscript especially in the introduction, abstract and conclusions sections. We also removed some technical details from the Results section. These changes (many of which are deletions) were not highlighted in yellow to make it easier to see the revisions that respond to specific questions of the reviewers.

2. Line 417. The hypothesis of a dynamic substrate recognition site is intriguing, but no experimental evidence (e.g., mutagenesis, binding assays, or NMR data) has been provided to confirm structural changes in solution. Since no inhibitor-bound structure was obtained, an alternative approach, such as molecular dynamics simulations, could be suggested to support the hypothesis.

RESPONSE: Although indeed no direct evidence was presented for the changes in the substrate binding site from the crystal structure with the pro-domain bound to the presumed structure when the cytolysin substrate peptides/inhibitors are bound, we presented several pieces of data that in our mind requires such a change. These include: (1) differences between the SAR studies and the XRD structure (especially the dependence on Asp -5 which the XRD structure does not predict), (2) perturbations in the SEC traces of inhibitor-treated CylA that suggests interactions with both pro-CylA and apo-CylA that change the retention times and hence that need to change the overall apparent size, (3) substoichiometric activation of CylA when treated with inhibitor, which is best explained by a non-inhibitory binding state that increases the activity, and (4) substrate activation of CylA. We did not do the suggested MD simulations (they would also only offer indirect support) but do now also refer to the AlphaFold 3 predicted binding pose of the substrate peptide (Supplementary Figure 15). This structure of CylA:GDVQAE shows a salt bridge between a Lys residue and Asp -5 that was absent in our XRD structure of pro-CylA. We feel that the hypothesis of dynamic substrate recognition is consistent with the experimental data. Unfortunately, thus far, apo-CylA or inhibitor-bound CylA have remained recalcitrant to crystallization. Importantly, in our view the proposed dynamic substrate recognition hypothesis is much less important than the demonstration that CylA can be inhibited resulting in abrogation of cytolysin production.

3. How can the authors rule out that factors other than cytolysin contribute to erythrocyte lysis in *E. faecalis*? It is possible that compound 9 also inhibits other, yet unknown, (virulence) factors as a side effect. To confirm that the observed lysis effects on erythrocytes are specifically due to cytolysin in the bacterial supernatants, including a cytolysin mutant as a control is essential. This mutant control is preferable to simply omitting bacteria from wells containing erythrocytes.

RESPONSE: As suggested, we repeated the solution-phase hemolysis experiment with previously reported mutants of C+ E. faecalis in a genetic background where cytolysin is known to be the sole hemolytic element [J. Bacteriol. 158, 777-783 (1984); J. Bacteriol. 176, 7335-7344 (1994)]. Consistent with previous findings, curing E. faecalis of the conjugative plasmid pADI, which bears the cytolysin BGC in WT C+E. faecalis [J. Bacteriol. 158, 777-783 (1984)], rendered the strain non-hemolytic. Introducing a mutation to CylA also abrogated hemolysis [J. Bacteriol. 176, 7335-7344 (1994)]. Treating the parent strain, which has WT hemolytic activity [J. Bacteriol. 158, 777-783 (1984)], with inhibitor 9 significantly reduced hemolysis. These results were very similar to those observed with E. faecalis ATCC 29212, which was run alongside these mutants. We have updated Fig. 8 with these results. We thank the reviewer for the suggestion, which has further solidified our conclusions.

Minor Points:

1. Line 178. The explanation that up to 30% of the deprotected peptide underwent protodeboronation is clear. However, it would be helpful to briefly discuss any implications this may have for the stability of the inhibitor.

RESPONSE: We added a brief discussion of the stability of alkylboronic acids after mentioning the protodeboronation. This process takes place under the acidic conditions of removing the protecting groups from the synthetic peptides. To more completely address the reviewer's comment, the stability of 9 to human serum and to C+ E. faecalis was also explored in new experiments. We tested the stability of 9 in the presence of human serum using an established protocol [Jenssen, H. & Aspino, S. I. in Peptide-Based Drug Design (ed Laszlo Otvos) 177-186 (Humana Press, 2008)]. The IC₅₀ of a sample of inhibitor 9 increased to 55 nM (from 8 nM) after a 3 h incubation in human serum at 37 °C. The IC₅₀ of a sample of inhibitor 9 increased to 36 nM after a 20 h incubation with C+ E. faecalis at 37 °C. See Supplementary Figure 19 in the revised version. These data show that the inhibitor has relatively good stability in these conditions.

2. Why was Fmoc-SPPS chosen over other methods, such as Boc/OBzl?

RESPONSE: Boc SPPS, although more rapid and often occurring with less side reactions, was not used because we do not have the specialized equipment necessary for Boc SPPS (which in its most commonly used form requires HF cleavage from the resin) and the sequences were not sufficiently challenging to warrant pursuing a Boc synthesis.

3. Line 236. The finding that compound 9 (cyclopropylglycine variant) was more potent than the parent inhibitor is interesting but is not deeply explored. Could this be due to altered conformational flexibility or enhanced binding interactions?

RESPONSE: Our hope was that conformational constriction of the Val side chain would limit the entropy penalty associated with tight binding and increase the exergonicity of the binding event. This was mentioned in part in the original manuscript, but we have expanded the text to make this clearer.

4. Consistency: warhead or war head

RESPONSE: Thank you for the suggestion. We have made this now consistent throughout the manuscript.

Reviewer #2 (Remarks to the Author):

Cytolysin is peptide produced by C+ *Enterococcus faecalis*. This *E. faecalis* strain is associated with significant mortality in patients with alcohol-induced hepatitis. A long and a short isoform of cytolysin is initially synthesized by the ribosome followed by subsequent posttranslational modifications involving sequential processing by the enzymes CylM, CylB and CylA. CylA is a secreted serine protease which cleaves a hexapeptide secretion signal with the sequence GDVQAE from both the long and short isoforms, thus producing the active cytolysins. The manuscript by Moreira et al. describes the development of boronic acid containing peptide inhibitors targeting CylA, which are derived from the hexapeptide leader sequence of the long and short cytolysin isoforms. The best peptide in which the valine at position 3 is replaced with a cyclopropylglycine exhibits an EC(50) values of ~10 nM, an ~3-fold reduction compared to the starting sequence. With selected inhibitors the authors performed detailed inhibition studies with the purified enzyme and also with C+ *E. faecalis* cells. Furthermore, the authors determined a high resolution crystal structure of the enzyme in which its pro-domain is still bound to the catalytic domain via its C-terminal tail. As the sequence of the C-terminal tail (PDFKPE) is related to the sequence of the peptide-based inhibitors, this structure provides insights how the inhibitors could be bound.

Overall, this is a very interesting study employing a variety of experimental techniques, which yielded high affinity starting compounds targeting an potential drug target. Listed below are points, which the authors should address in a potential revised version.

RESPONSE: Thank you for the positive overall evaluation. We have made the revisions indicated below in response to the reviewer suggestions.

1. A) Why is the intrinsic activity of the enzyme to cleave especially mCylL(L) but also mCylL(S) so small (Fig. 3).

RESPONSE: Thank you for the question which illustrates we need to explain the conditions better. This experiment was conducted at an enzyme concentration of 4 nM and a substrate concentration of 10 μM for just 30 min (conditions now added to the text). Conversion was purposely limited because we felt that having substrate present in all of the traces would make the resulting figure more visually compelling and it would better illustrate the potency since complete inhibition is observed at high substrate concentrations.

B) Does the conversion achieve 100% after longer incubation times or incubation at higher temperatures?

RESPONSE: Yes, the conversion under these conditions approached completion if left for longer periods of time. We reported using similar conditions to prepare cytolysin biosynthetically [ACS Infect. Dis. 7, 2445-2454 (2021)]. We briefly mention this in the new legend to Fig. 3.

C) Why is there even such a pronounced difference between the long and short forms? This is rather surprising in light of the conservation of the two cytolysins around the cleavage site.

RESPONSE: Thank you for the question. Previously, CylA was shown to process mCylL_L and mCylL_S faster than the unmodified full-length precursors suggesting that the enzyme may recognize residues after the cleavage site [J. Ind. Microbiol. Biotechnol. 46, 537-549 (2019)]. It was also found that CylA processed mCylL_S faster than mCylL_L, which as the reviewer mentions is also evident in Fig 3 of this manuscript. For completeness, we added a brief discussion to the methods section. We did not want to complicate the main text as this aspect is not directly related to the inhibition study and has been reported previously.

2. In the present form Fig. 4 is helpful to illustrate how the assay works, however, what one would really like to see instead would be the actual data. Please modify the figure so that the existing Fig. 4 becomes panel (A) of a revised Fig. 4 and in subsequent panels show the experimental data as presented in the current Fig. S3 for a subset of inhibitors. Data for all remaining inhibitors should be shown in a revised version of Fig. S3.

RESPONSE: Thank you for the suggestion. Fig. S3 in the previously submitted version is now incorporated into Fig. 4 in the revised version. Liposome permeabilization curves and dose-response curves for all inhibitors were added to the new Supplementary Fig. S3.

3. The aSEC data shown in Fig. 4D are rather puzzling, especially the shift of the two peaks. Assuming that the peak at ~11.4 min shifts to the right because the complex increases in size due to inhibitor binding, why is there a complete shift already at substoichiometric inhibitor concentrations? If it is not inhibitor binding, what causes this shift? For the peak at 11.8 min, why is there a shift to later elution times? Please provide explanations for these effects. Ideally, the aSEC data should be replaced with SEC-MALLS data if the authors can gain access to such an instrument. At the least, the authors should show the full chromatograms (maybe in the supplement) and also analyze the corresponding fractions on SDS-PAGE gels.

RESPONSE: In general, peaks shift to the left (shorter retention times) when size increases; we have added labels to Fig. 5D and E to make this clearer. These labels are from analysis of an external standard (see Supplementary Figure 7 in the revised version). We have also added a label for the CylA:inhibitor complex.

To address the reviewer's specific comments, we have added full length chromatograms of all SEC traces to the SI (see Supplementary Figure 7 in the revised version) and added SDS-PAGE analysis of the fractions collected during aSEC of CylA (see Supplementary Figure 8), which support our assignment of the peaks but do not provide insights as to why the peaks shift. Unfortunately, we do not have an SEC-MALS that is appropriate for analyzing this system.

Regarding the questions about interpretation of the observed results, we are reluctant to add any speculations in the absence of direct structural data. SEC separates proteins based on their radius of gyration in solution. The peak at 11.4 min shifting to an earlier elution time suggests interactions between pro-CylA and the inhibitor increase the radius of gyration of the enzyme. This could be interpreted as the inhibitor increasing the flexibility of the enzyme which is consistent with it entering a more dynamic state favoring pro-domain ejection. Seeing this shift already at substoichiometric amounts of inhibitor is similar to the activation at substoichiometric amounts of inhibitor and suggests a catalytic induction of a new state. When the inhibitor binds to CylA-96-412, the resulting complex has a longer retention time, consistent with the enzyme contracting upon inhibitor binding. This could be interpreted as the inhibitor stabilizing the enzyme. But as noted, we are reluctant to make claims in the paper since we have no direct evidence. We present the data and readers can make interpretations or the data can be used in future investigations. We feel presenting them contributes to the study, but prefer not to add speculative statements.

4. There are a few points regarding the structural studies.

(A) Although not stated in the legend to Fig. 6B, this is probably a 2Fo-Fc map, which should be replaced with an omit map.

RESPONSE: The map shown is an unbiased omit map (Fo-Fc) generated using phases from the final structure, without the inclusion of residues P90 to E9, after rounds of simulated annealing refinement. The description of the map calculation is now included in the corresponding figure legend for clarity.

(B) Please also show Ser359 in panel B to make it easier for the reader to see where the peptide is relative to the active site cysteine.

RESPONSE: We agree and thank the reviewer for this suggestion. Apologies for omitting the label for this residue originally. Figure 6B has been revised and Ser359 is now labelled.

(C) Based on the experimental structure please provide a model with the bound inhibitor so that the reader can judge the position of the boronic acid with respect to the active site serine.

RESPONSE: The suggested change was made. This new figure is now included as Supplemental Figure 14.

(D) Please describe and show (in a separate panel in Fig. 6) the interactions between the pro-domain and catalytic domain. What is the amount of buried surface in the interface?

RESPONSE: The suggested change was made. Due to the complexity of the figure showing the interaction, we have opted to add it as separate Supplemental Figures 12 (space filling) and 13 (LigPlot). A discussion of the interactions interface (and buried surface area) has been added.

(E) Please replace all ColabFold predictions with Alphafold3 predictions, as this represents the current state of the art.

RESPONSE: The ColabFold structure featured in Supplementary Fig. 9 was replaced with a structure of CylA:GDVQAE predicted by AlphaFold 3. The two structures are very similar, and the conclusions drawn from the ColabFold structure apply to the AlphaFold 3 structure. See Supplementary Figure 15 in the revised version.

(F) I find Fig. 7 not very helpful to convey the similarities in the pro-domains. Please show a superposition of the three structures (probably in what in Pymol is called a ribbon representation). If needed, the cartoon representations could be shown at the same scale next to the superimposition.

RESPONSE: The suggested change was made. This new figure is now included as Supplemental Figure 16.

(G) Given the high resolution the R-factors of the refined model are somewhat disappointing. Did the authors try to refine individual anisotropic B-factors?

RESPONSE: We have refined individual anisotropic B-factors. For comparison purposes, a survey of 20 structures that have been deposited in the RCSB Data Bank in the last 2 years, determined at resolutions between 1.0 to 1.3 Å, have a range of free R factors between 0.168 to 0.214. We consider the free R factor of our structure (0.186) to be well within this range.

(H) The authors should comment on the low completeness of the data in the highest resolution shell, which is probably caused by the detector distance being too large. Given the data collection statistics (R(pim), I/sig/ and CC(1/2)) the data extended apparently to even higher resolution. The authors need to comment on these aspects.

RESPONSE: We now provide greater details of data resolution limitation in the Methods section.

(I) The section in Materials and Method should probably be entitled "Crystal structure analysis of His(6)-CalA-27-412" and not "Crystallization and Data Collection of ..." since it also covers the structure solution, model building and refinement steps.

RESPONSE: We agree. The suggested change was made.

5. Please provide a better explanation for the data presented in Fig. 8C. Shouldn't the concentration of CylL(S) decrease in the presence of inhibitor and given the data presented in Fig. 8B this should occur for sure at a concentration of 10 μM ? This unexpected behavior is referred to in the text as "bidirectional modulation" (line 500), which seems rather vague.

RESPONSE: We removed the term bidirectional modulation and instead included a more complete explanation of the data in Fig. 8.

The data in Figure 8B suggested that the inhibitor may be ineffective at 10 μM which is consistent with the data presented in Figure 8C. We speculated that the increase in CylL_S concentration detected by LCMS when cells were treated with 10 μM of inhibitor could be the result of activation of CylA by the inhibitor. This phenomenon was seen in Figs. 5C-E and Supplementary Fig. 6 and this explanation was provided in the original version of the manuscript. However, other explanations of similar validity can be surmised. For example, it is known that CylL_S is a better substrate for CylA than CylL_L [J. Ind. Microbiol. Biotechnol. 46, 537-549 (2019)] and thus imbalances favoring CylL_S maybe be observed when CylA is partially inhibited. These imbalances may be compounded by the co-operative aggregation of CylL_S and CylL_L resulting in increases in extractable CylL_S that are not reflective of the rate of toxin maturation. We have added this alternative explanation to the manuscript and contextualized the efficacy of 9 under these conditions by comparing our results to another study that used similar assay conditions [Sci. Rep. 8, 14578 (2018)].

Additional points:

Line 43: Replace "colon-rectal" with "colorectal"

RESPONSE: We agree. The suggested change was made.

Line 731: Please indicate the instrument used for these experiments.

RESPONSE: The suggested change was made. Lines 729-730, 750-751, and 758-759 state the instrument that was used in these experiments.

Fig. S1 and S2: Please add calculated masses of the inhibitors next to the chemical structures. Although the values are listed in Table S1, it will make it easier for the reader if the info is also included in the figure.

RESPONSE: The theoretical m/z values for the base peaks have been added to Fig. S1 and S2.

Table S3: Given the high resolution it seems justified to state the unit cell dimensions with two digits after the decimal point.

RESPONSE: We agree. The suggested change was made.

Fig. S6: A) Please explain "vi" and "vo" and state that in the absence of inhibitor "vi/vo" is equal to one.

RESPONSE: An explanation of vi and vo has been added to the description of this figure. See Supplementary figure 6 in the revised version.

B) Why are error bars sometimes missing?

RESPONSE: Thank you for the question. The error was miscalculated for the run that used mCylL_S at a concentration of 80 nM. We apologize for overlooking this and have updated the plot. The runs that used a concentration of mCylL_S equal to 3 μM proceeded with a standard deviation of 9 E -4 so the width of this error bar was smaller than the width of the outline of the box. To make this plot clearer, we have changed it to include data points.

Fig. S7: The same reservations described for Fig. 5D apply here, i.e. a clear assignment of the peaks is missing. In addition, the whole chromatogram should be shown.

RESPONSE: Figure S7 was changed to include whole chromatograms. These chromatograms are now labelled similar to those in Fig 5D.

Fig. S9: Repeat prediction with Alphafold3 plus add prediction of crystal structure to get an idea about the accuracy of the prediction when compared to the experimentally derived structure.

RESPONSE: The suggested change was made. This new figure is now included as Supplementary Figure 11.

Fig. S11: Where is the curve for the negative control?

RESPONSE: The responses from the negative control wells (vehicle) were used to calculate % survival using the following equation:

$$\% \text{ survival} = 100\% - \frac{(\text{Absorbance of Treated well} - \text{Avg}(\text{Absorbance of Negative control}))}{(\text{Avg}(\text{Absorbance of Positive control}) - \text{Avg}(\text{Absorbance of Negative control}))} * 100\%$$

We included a description of this calculation to the revised manuscript. Based on this equation, the responses produced by negative control wells are defined as having a value of 100% so including them doesn't provide the reader with any new information. For this reason, these values were omitted.

Fig.S12 and S13: The legends are missing.

RESPONSE: Legend was added.

We thank the reviewer for the careful read and the suggestions that have improved our manuscript.

Reviewer #3 (Remarks to the Author):

This is an excellent study. The authors are addressing a problem of key relevance to chemical biologists and microbiologists seeking to combat cytolysin virulence and develop treatments for cytolysin-induced disease states. Based on the detailed studies of the biosynthesis of the two-component lantibiotic cytolysin previously published by this group, this work identifies the extracellular serine protease CylA as a suitable target for preventing maturation of both of the peptide components of cytolysin. A series of peptide boronic acids were designed and synthesised based on the sequences recognised by CylA, enabling identification of nanomolar inhibitors. A thorough study of the inhibition of CylA by inhibitor 9 is presented. Importantly, the authors have succeeded in obtaining an X-ray crystal structure of CylA bound to the proteolytically excised pro-domain, which provided insights into the enzyme-inhibitor interactions. The authors have provided well-written and comprehensive experimental details and characterisation data in the Experimental and in the Supporting Information.

The manuscript is well written and will be of interest to readers of Nature Communications. I recommend publication, subject to the authors addressing some minor points and queries:

There is an ambiguity in Figure 2. The synthesis of peptides 1b, 3b - 9b are shown as starting from 2-ClTrt resin pre-loaded with Fmoc-L-Ala. However in peptide 6 this residue is Abu (and therefore this was presumably the residue pre-loaded on the resin)

RESPONSE: We agree and changed the figure to clarify this.

The characterization of the peptides is good (Supplementary Figures 1 and 2). To aid reproducibility, could the HPLC retention times for the purified peptides be included?

RESPONSE: We agree. Retention times are now included in Supplementary Table 2. We also included more detailed information on our LC separations into the material and methods sections.

The rationale for using mCylL_S and mCylL_L as substrates for CylA is not made clear. As the in vivo substrates for CylA are CylL_L' and CylL_S', would it not have been more relevant to use these?

RESPONSE: Thank you for the question as we should have made this more clear. We decided to use mCylL_S and mCylL_L over the natural substrate for several reasons. First, CylA readily accepts these substrates making them suitable replacements for the natural substrates in activity assays [J. Ind. Microbiol. Biotechnol. 46, 537-549 (2019)]. Second, mCylL_S and mCylL_L are easier to obtain in the amounts required for these experiments because they do not aggregate like CylL_S' and CylL_L' and they do not need to be treated with a membrane-bound protease (CylB) that has not been reconstituted. Third, the enhanced solubility of mCylL_S and mCylL_L in aqueous solution improved the activity assays. We have added a statement to this effect in the revised version.

In the discussion of Figure 5D, the authors mention that several overlapping peaks are observed by aSEC when His6-CylA-27-412 is analysed in the presence of inhibitor 9. Could native ESI mass spectrometry have been useful here to identify both complexation and folding/unfolding?

RESPONSE: Thank you for the suggestion. We now tried to characterize the CylA:inhibitor complex by native ESI-MS but signals corresponding to CylA could not be observed in the mass spectrum, possibly because of the aggregation tendency of the protein. Instead, we added SDS-PAGE analysis of the fractions to the Supplementary Information (Supplementary Fig. 8), which agrees with our interpretation of the peaks.